# Feed-Forward Bullet-Time Reconstruction of Dynamic Scenes from Monocular Videos

**Hanxue Liang**[1,2]*, **Jiawei Ren**[1,3]*, **Ashkan Mirzaei**[1,4]*, **Antonio Torralba**[1,5],
**Ziwei Liu**[3], **Igor Gilitschenski**[4], **Sanja Fidler**[1,4,6],
**Cengiz Oztireli**[2], **Huan Ling**[1,4,6], **Zan Gojcic**[1]†, **Jiahui Huang**[1]†

[1]NVIDIA, [2]University of Cambridge, [3]Nanyang Technological University,
[4]University of Toronto, [5]MIT, [6]Vector Institute
https://research.nvidia.com/labs/toronto-ai/bullet-timer/

## Abstract

Recent advancements in static feed-forward scene reconstruction have demonstrated significant progress in high-quality novel view synthesis. However, these models often struggle with generalizability across diverse environments and fail to effectively handle dynamic content. We present BTimer (short for BulletTimer), **the first motion-aware feed-forward model for real-time reconstruction and novel view synthesis of dynamic scenes**. Our approach reconstructs the full scene in a 3D Gaussian Splatting representation at a given target ('bullet') timestamp by *aggregating* information from all the context frames. Such a formulation allows BTimer to gain scalability and generalization by leveraging both static and dynamic scene datasets. Given a casual monocular dynamic video, BTimer reconstructs a *bullet-time*[1] scene within 150ms on $256 \times 256$ resolution while reaching state-of-the-art performance on both static and dynamic scene datasets, even compared with optimization-based approaches.

## 1 Introduction

Multi-view reconstruction and novel-view synthesis are long-standing challenges in computer vision, with numerous applications ranging from AR/VR to simulation and content creation. While significant progress has been made in reconstructing static scenes, dynamic scene reconstruction from monocular videos remains challenging due to the inherently ill-posed nature of reasoning about dynamics from limited observations [2].

Current methods for static scene reconstruction can be broadly divided into two categories: optimization-based [3, 4] and feed-forward [5, 6] approaches. However, extending

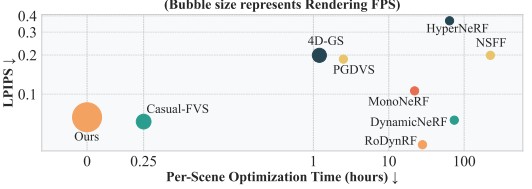

Figure 1: **Rendering quality vs. speed.** Our model can reconstruct and render dynamic scenes at a much faster speed than existing approaches with a competitive quality. Numbers are reported on NVIDIA Dynamic Scene Dataset [1]

both of these to *dynamic scenes* is not straightforward. To reduce the ambiguities of scene dynamics, many optimization-based methods aim to constraint the problem with data priors such as depth and

---

*/†: Equal contribution/advising.

[1]In this paper, we define *bullet-time* as the instantiation of a 3D scene *frozen* at a given/fixed timestamp $t$.

39th Conference on Neural Information Processing Systems (NeurIPS 2025).

optical flow [2, 7, 8, 9]. However, balancing these priors with the data remains challenging [10, 11]. Moreover, per-scene optimization is time-consuming and thus difficult to scale.

On the other hand, to avoid the lengthy per-scene-optimization, recent feed-forward approaches [12, 13, 14, 15, 16, 5] explored learning generalizable models on large-scale datasets to directly perform static scene reconstructions, thereby learning strong priors from data. These inherent priors could help resolve ambiguities due to complex motion, but none of previous approaches have yet been extended to dynamic scenes. This limitation stems from both the complexity of modeling dynamic scenes and the lack of 4D supervision data. The only feed-forward dynamic reconstruction model [17] is thus trained on synthetic object-centric datasets, requires fixed camera viewpoints and multiview supervision, and cannot generalize to real-world scene scenarios.

In this work, we aim to answer the question: *How can one build a feed-forward reconstruction model that can handle dynamic scenes effectively?* We build upon the recent success of the pixel-aligned 3D Gaussian Splatting (3DGS [18]) prediction models [5] and propose a novel *bullet-time* formulation for feed-forward dynamic reconstruction. The core idea is simple yet effective: we add a *bullet-time* embedding to the context (input) frames, indicating the desired timestamp for the output 3DGS representation. Our model is trained to aggregate the predictions of context frames to reflect the scenes at the *bullet* timestamp, yielding a spatially complete 3DGS scene. This design not only naturally unifies the static and dynamic reconstruction scenarios, but also enables our model to become implicitly motion-aware while learning to capture scene dynamics. In particular, the proposed formulation **(i)** allows us to pre-train our model on large amounts of *static* scene data, **(ii)** scales effectively across datasets, without being constrained by duration and frame rates of input videos, and **(iii)** outputs volumetric video representations that inherently support multiple viewpoints. Meanwhile, in the presence of fast motions, we additionally introduce a Novel Time Enhancer (NTE) module to predict the intermediate frames before feeding them to the main model.

In summary, we present BTimer, *the first feed-forward model for real-time reconstruction and novel view synthesis of dynamic scenes.* To achieve this goal, we introduce the core bullet-time formulation and develop a curriculum training strategy that enables the learning of a highly generalizable model on a large, carefully curated dataset comprising both static and dynamic scenes. Furthermore, we present an additional NTE module to effectively handle fast motions, enhancing the model's robustness in challenging scenarios. Our method is highly efficient: feed-forward inference with 12 context frames of $256 \times 256$ resolution only costs 150ms on a single GPU, and the output 3DGS can be rendered in real-time. BTimer is capable of handling both static and dynamic reconstructions. It achieves competitive results on various reconstruction benchmarks, even surpassing many expensive per-scene optimization-based methods, as illustrated in Fig. 1.

## 2 Related work

**Dynamic 3D representations.** Depending on the tasks at hand, typical choices of 3D representations include voxels [19, 20], implicit fields/NeRFs [21, 3, 22], and point clouds/3D Gaussians [23, 18]. Representing dynamics on top has an even larger design space: One existing line of works directly builds a '4D' representation to enable feature queries at arbitrary positions and timestamps from an implicit field [24, 25] or via marginalization at a given step [26, 27], with the extensibility to higher dimensions such as material [28]. Another line of work first defines a canonical 3D space, and learns a deformation field to warp the canonical space to the target frame. While these methods learn additional information about shape correspondences, their performance heavily relies on the quality and topology of the canonical space.

**Dynamic novel view synthesis.** For tasks that require a relatively smaller view extrapolation, the problem of novel view synthesis can be tackled without explicit 3D geometry in the loop, using depth warping [1] or multi-plane images [29]. Otherwise, the study of novel view synthesis of dynamic scenes [30, 4] is mainly on (1) effectively optimizing the 3D representation through input images through monocular cues [31, 8, 11, 10] or geometry regularizations [32, 33], and (2) being able to render fast with grids [34], local-planes [35], or dynamic 3D Gaussians [36] formulation. Our method aims to provide a dynamic representation that is fast to build within hundreds of milliseconds while reaching competitive rendering quality as the above optimization-based methods.

**Feed-forward reconstruction models.** In many applications where the reconstruction speed is crucial, most optimization-based reconstruction methods become less preferable. To this end, one line

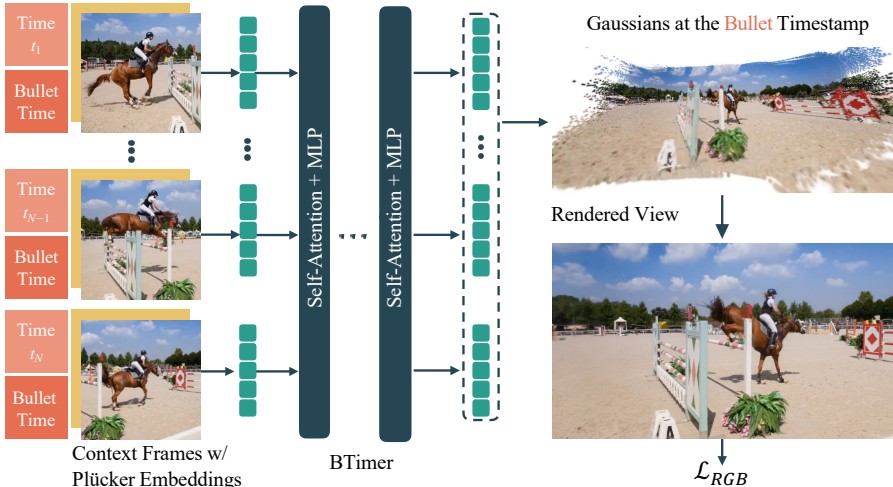

Figure 2: **BTimer.** The model takes as input a sequence of context frames and their Plücker embeddings, along with the context timestamp and target ('bullet') timestamp embeddings. It then directly predicts the 3DGS representation at the bullet timestamp.

of work that starts to emerge is fully feed-forward models that directly regress from 2D images to 3D, represented as either neural field [13, 37], triplanes [12], 3D Gaussians [14, 5, 15], sparse voxels [38], or latent tokens [16]. Crucially, while feed-forward reconstruction models for static scenes have seen development, the extension to dynamic scenes is still challenging. Existing methods either require hard-to-acquire consistent video depth as input [39], do not support rendering [40], or only work on object-scale data [17]. In contrast, our method supports reconstructing from a monocular video containing dynamic scenes in a fully feed-forward manner, and is able to render at arbitrary viewpoints and timestamps.

## 3   Method

**Overview.** Given a monocular video (image sequence) represented by $\mathcal{I} = \{\mathbf{I}_i \in \mathbb{R}^{H \times W \times 3}\}_{i=1}^N$ with $N$ frames of width $W$ and height $H$, along with known camera poses $\mathcal{P} = \{\mathbf{P}_i \in \mathbb{SE}(3)\}_{i=1}^N$, intrinsics, and corresponding timestamps $\mathcal{T} = \{t_i \in \mathbb{R}\}_{i=1}^N$, our goal is to build a feed-forward model capable of rendering high-quality novel views at arbitrary timestamps $t \in [t_1, t_N]$.

The core of our approach is a transformer-based *bullet-time* reconstruction model, named **BTimer**, that takes in a subset of frames $\mathcal{I}_c \subset \mathcal{I}$ (denoted as *context frames*) along with their corresponding poses $\mathcal{P}_c \subset \mathcal{P}$ and timestamps $\mathcal{T}_c \subset \mathcal{T}$, and outputs a complete 3DGS [18] scene frozen at a specified *bullet* timestamp $t_b \in [\min_{\mathcal{T}_c}, \max_{\mathcal{T}_c}]$ (§ 3.1). Iterating over all $t_b \in \mathcal{T}$ results in a full video reconstruction represented by a sequence of 3DGS. We further introduce a Novel Time Enhancer (NTE) module that synthesizes interpolated frames with timestamps $t \notin \mathcal{T}$ (§ 3.2). The output of the NTE module is used along with other context views as input to the bullet-time model to enhance reconstruction at arbitrary intermediate timestamps. To effectively train our model, we carefully design a learning curriculum (§ 3.3) that incorporates a large mixture of datasets containing both static and dynamic scenes, to enhance motion awareness and temporal consistency of our models.

### 3.1   BTimer reconstruction model

**Model design.** Inspired by [5], our BTimer model uses a ViT-based [41] network as its backbone, consisting of 24 self-attention blocks with LayerNorms [42] applied at both the beginning and the end of the model. We divide each input context frame $\mathbf{I}_i \in \mathcal{I}_c$ into $8 \times 8$ patches, which are projected into feature space $\{\mathbf{f}_{ij}^{\text{rgb}}\}_{j=1}^{HW/64}$ using a linear embedding layer. The camera Plücker embeddings [43] derived from the camera poses $\mathbf{P}_i \in \mathcal{P}_c$ and the time embeddings (detailed later) are processed similarly to form the camera pose features $\{\mathbf{f}_{ij}^{\text{pose}}\}$ and the time features $\{\mathbf{f}_i^{\text{time}}\}$ (shared for all patches $j$). These features are added together to form the input tokens for the patches of the context

frame $\{\mathbf{f}_{ij}\}_{j=1}^{HW/64}$, where $\mathbf{f}_{ij} = \mathbf{f}_{ij}^{\text{rgb}} + \mathbf{f}_{ij}^{\text{pose}} + \mathbf{f}_i^{\text{time}}$. The input tokens from all context frames are concatenated and fed into the Transformer blocks.

Each corresponding output token $\mathbf{f}_{ij}^{\text{out}}$ is decoded into 3DGS parameters $\mathbf{G}_{ij} \in \mathbb{R}^{8 \times 8 \times 12}$ using a single linear layer. Each 3D Gaussian is paramaterized by its RGB color $\mathbf{c} \in \mathbb{R}^3$, scale $\mathbf{s} \in \mathbb{R}^3$, rotation represented as unit quaternion $\mathbf{q} \in \mathbb{R}^4$, opacity $\sigma \in \mathbb{R}$, and ray distance $\tau \in \mathbb{R}$, resulting in 12 paramaters per Gaussian. The 3D position of each Gaussian $\boldsymbol{\mu} \in \mathbb{R}^3$ is obtained through pixel-aligned unprojection as $\boldsymbol{\mu} = \mathbf{o} + \tau \mathbf{d}$, where $\mathbf{o} \in \mathbb{R}^3$ and $\mathbf{d} \in \mathbb{R}^3$ are the ray origin and direction obtained from $\mathbf{P}_i$.

**Time embeddings.** The aforementioned input time feature $\mathbf{f}_i^{\text{time}}$ is obtained from: **(i) context** timestamp $t_i$ that is separate for each context frame $\mathbf{I}_i$, and **(ii) bullet** timestamp $t_b$ that is shared across all context frames $i$. Both timestamp scalars are encoded using standard Positional Encoding (PE) [44] with sinusoidal functions, and then passed through two linear layers to obtain the features $\mathbf{f}_i^{\text{ctx}}$ and $\mathbf{f}_i^{\text{bullet}}$ respectively. Finally, we set $\mathbf{f}_i^{\text{time}} = \mathbf{f}_i^{\text{ctx}} + \mathbf{f}_i^{\text{bullet}}$.

**Supervision loss.** Our model is supervised only by losses defined in the RGB image space, bypassing the need for any source of 3D ground truth that is hard to obtain for real data. The final loss is a weighted sum of Mean Squared Error (MSE) loss and Learned Perceptual Image Patch Similarity (LPIPS) [45] loss between the images rendered from the 3DGS output and the ground-truth image:

$$\mathcal{L}_{\text{RGB}} = \mathcal{L}_{\text{MSE}} + \lambda \mathcal{L}_{\text{LPIPS}}, \quad (1)$$

with $\lambda = 0.5$.

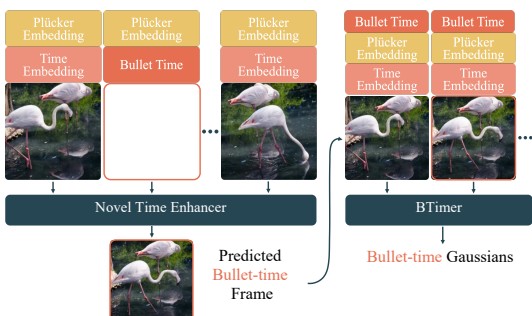

Figure 3: **NTE Module.** It takes as input the target bullet time embedding, target pose, as well as adjacent frames to directly predict corresponding RGB values. The predicted frame is then used in BTimer as *bullet* frame for novel time reconstruction.

Careful selection of input context frames and corresponding supervision frames (at the bullet timestamp) during training is essential for stable training and good convergence. In practice, we find the combination of the following two strategies particularly effective: **(i) In-context Supervision** where the supervision timestamp is randomly selected from the context frames, encouraging the model to accurately localize and reconstruct the context timestamps. For multi-view video datasets, images from additional viewpoints can also contribute to the loss. **(ii) Interpolation Supervision** where the supervision timestamp lies between two adjacent context frames. This forces the model to interpolate the dynamic parts while maintaining consistency for the static regions. The interpolation supervision significantly impacts our final performance (*cf.* § 4.4 for details); without it, the model falls into a local minima by positioning the 3D Gaussians close to the context views but hidden from other views.

**Inference.** Our *bullet-time* formulation makes it straightforward to reconstruct a full video, which only involves iteratively setting the bullet timestamp $t_b$ to every single timestamp in the video, and can be done efficiently in parallel. For a video longer than the number of training context views $|\mathcal{I}_c|$, at timestamp $t$, apart from including this exact timestamp and setting $t_b = t$, we uniformly distribute the remaining $|\mathcal{I}_c| - 1$ required context frames across the whole duration of the video to form the input batch with $|\mathcal{I}_c|$ frames.

## 3.2 Novel time enhancer (NTE) module

While our BTimer model can already reconstruct the 3DGS representation for all observed timestamps, we notice that forcing it to reconstruct at a novel intermediate timestamp, *i.e.* performing interpolation at $t_b \notin \mathcal{T}$, leads to suboptimal results. In such cases, the exact bullet-time frame cannot be included in the context frames as it does not exist. Our model specifically fails to predict a smooth transition between adjacent video frames when the motion is complex and fast. This is mainly caused by the inductive bias of pixel-aligned 3D Gaussian prediction. To mitigate this issue, we propose a *3D-free* Novel Time Enhancer (NTE) module that directly outputs images at given timestamps, which are then used as input to our BTimer model, as illustrated in Fig. 3.

**NTE module design.** The design of this module is largely inspired by the very recent decoder-only LVSM [16] model. Specifically, NTE copies the same ViT architecture from the BTimer model, but the time features of input context tokens only encode their corresponding context timestamps (*i.e.* we set $\mathbf{f}_i^{\text{time}} = \mathbf{f}_i^{\text{ctx}}$). Additionally, we concatenate extra target tokens to the input tokens, which encode the target timestamp and the target pose for which we want to generate the RGB image. Following [16], we use QK-norm to stabilize training. Implementation-wise we apply an attention mask that masks all the attention to the target tokens, so KV-Cache (*cf.* [46]) can be used for faster inference. From the output of the Transformer backbone, we only retain the target tokens, which we then unpatchify and project to RGB values at the original image resolution using a single linear layer. The interpolation model is trained with the same objective as the main BTimer model (see § 3.1), but the output image is directly decoded from the network and not rendered from a 3DGS representation.

**Integration with BTimer.** While the NTE module can be used on its own to generate novel views, we empirically find the novel-*view*-synthesis quality to be inferior (§ 4.4). We hence propose to integrate it with our main BTimer model. To reconstruct a bullet-time 3DGS at $t_b \notin \mathcal{T}$, we first use NTE to synthesize $\mathbf{I}_b$ at the timestamp $t_b$, where the target pose $\mathbf{P}_b$ is linearly interpolated from the nearby context poses in $\mathcal{P}$, and the context frames are chosen as the nearest frames to $t_b$. To accelerate the inference of the interpolation model, we use the KV-Cache strategy. In practice we observe that the interpolation model adds negligible overhead to the overall runtime.

## 3.3 Curriculum training at scale

One important lesson people have learned from training deep neural networks is to scale up the training [47, 48], and the model's generalizability is largely determined by the data diversity. Since our bullet-time reconstruction formulation naturally supports both static (by equalizing all elements in $\mathcal{T}$) and dynamic scenes, and requires only RGB loss for weak supervision, we unlock the potential of leveraging the availability of numerous static datasets to pretrain our model. We hence aim to train a *kitchen-sink* reconstruction model that is *not specific* to any dataset, making it generalizable to both static and dynamic scenes, and capable of handling objects as well as both indoor and outdoor scenes. This is in contrast to, *e.g.*, GS-LRM [5] or MVSplat [14] where one needs different models in different domains.

Notably, we apply the following training curriculum to BTimer and the NTE module separately, but during inference they are used jointly as explained in § 3.2.

**Stage 1: Low-res to high-res static pretraining.** To obtain a more generalizable 3D prior as initialization, we first pretrain the model with a mixture of *static* datasets. Time embedding will not be used in this stage. The collection of datasets covers object-centric (Objaverse [49]) and indoor/outdoor scenes (RE10K [50], MVImgNet [51], DL3DV [52]). The datasets cover both the synthetic and real-world domains and consist of 390K training samples. We normalize the scales of different datasets to be bounded roughly in a $10^3$ cube. Due to the complex data distribution, our training starts from a low-resolution few-view setting that reconstructs on $128 \times 128$ resolution from $|\mathcal{I}_c| = 4$ context views. To further increase the reconstruction details, we fine-tune the model from $128 \times 128$ by first increasing the image resolution to $256 \times 256$, and then fine-tune to $512 \times 512$.

**Stage 2: dynamic scene co-training.** After the training on static scenes, we start fine-tuning the model along with time embedding projection layers on dynamic scenes with available 4D data that contains monocular or multi-view synchronized videos. We leverage Kubric [53], PointOdyssey [54], DynamicReplica [55] and Spring [56] datasets for training. Due to the scarcity of 4D data, during this stage we keep the static datasets for co-training which provides more multi-view supervision and stabilizes the training. Additionally, we build a customized pipeline to label the camera poses from Internet videos (detailed below), and add them to our training set to further enhance the model's robustness towards real-world data.

**Stage 3: long-context window fine-tuning.** Including more context frames is vital when reconstructing long videos. Therefore, as a final stage, we increase the number of context views from $|\mathcal{I}_c| = 4$ to $|\mathcal{I}_c| = 12$ to cover more frames. Note that this stage does not apply to NTE as it only takes nearby frames as input.

**Annotating internet videos.** We randomly select a subset from the PANDA-70M [57] dataset, and cut the videos into short clips with $\sim 20\,\text{s}$ duration. We mask out the dynamic objects in the videos with Segment Anything Model [58] and then apply DROID-SLAM [59] to estimate the camera poses.

| Model | Rec. Time | PSNR↑ | SSIM↑ | LPIPS↓ |
|---|---|---|---|---|
| TiNeuVox [63] | 0.75 h | 14.03 | 0.502 | 0.538 |
| NSFF [2] | 24 h | 15.46 | 0.551 | 0.396 |
| T-NeRF [64] | 12 h | 16.96 | 0.577 | 0.379 |
| Nerfies [32] | 24 h | 16.45 | 0.570 | 0.339 |
| HyperNeRF [4] | 72 h | 16.81 | 0.569 | 0.332 |
| PGDVS [39] | 3 h† | 15.88 | 0.548 | 0.340 |
| Depth Warp | – | 7.81 | 0.201 | 0.678 |
| BTimer (**Ours**) | 0.98 s | 16.52 | 0.570 | 0.338 |

(a)

| Model | Rec. Time | Render FPS | PSNR↑ | LPIPS↓ |
|---|---|---|---|---|
| HyperNeRF [4] | 64 h | 0.40 | 17.60 | 0.367 |
| DynNeRF [65] | 74 h | 0.05 | 26.10 | 0.082 |
| RoDynRF [33] | 28 h | 0.42 | 25.89 | 0.065 |
| 4D-GS [66] | 1.2 h | 44 | 21.45 | 0.199 |
| Casual-FVS [9] | 0.25 h | 48 | 24.57 | 0.081 |
| PGDVS [39] | 3 h† | 0.70 | 24.41 | 0.186 |
| Depth Warp | – | – | 12.63 | 0.564 |
| BTimer (**Ours**) | 0.78 s | 115 | 25.82 | 0.086 |

(b)

Table 1: **Quantitative comparisons on dynamic datasets.** (a) DyCheck iPhone dataset [64] comparison. (b) NVIDIA Dynamic Scene dataset [1] comparison. The results are rendered on $480 \times 270$ resolution. 'Rec. Time' is per-scene reconstruction time. †: Video-consistent depth estimation step included. We highlight the best , second best , and third best results.

Low-quality videos or annotated poses are filtered out by measuring the reprojection error. The final dataset contains more than 40K clips with high-quality camera trajectories.

# 4 Experiments

In this section we first introduce necessary implementation details in § 4.1. We evaluate the performance of BTimer extensively on available dynamic scene benchmarks § 4.2, and demonstrate its *backward* compatibility with static scenes § 4.3. Ablation studies are found in § 4.4.

## 4.1 Implementation details

**Training.** Our backbone Transformer network is implemented efficiently with FlashAttention-3 [60] and FlexAttention [61]. We use gsplat [62] for robust and scalable 3DGS rasterization since the total number of 3D Gaussians generated by our model can be very large. For BTimer, the numbers of training iterations are fixed to 90K, 90K, and 50K for **Stage 1** training on $128^2$, $256^2$, and $512^2$ resolutions, and are 10K and 5K for **Stage 2** and **Stage 3** dynamic scene training respectively. We use the initial learning rates of $4 \times 10^{-4}$, $2 \times 10^{-4}$ and $1 \times 10^{-4}$ for the three stages, and apply a cosine annealing schedule to smoothly decay the learning rate to zero. Training is conducted on 32 NVIDIA A100 GPUs. The learning rate, training GPU numbers and training schedules mainly follow [16, 5]. Training cost analysis and ablation on batch size can be found in the Supplement. We use the same training strategy for NTE. The numbers of iterations are 140K, 60K, and 30K for the progressive training in **Stage 1**, and are 20K for **Stage 2**, with the same learning rate schedule as above. As introduced in § 3.3, we use a mixture of multiple datasets for training [49, 51, 50, 52, 53, 54, 55, 56] along with our 40K annotated dataset on PANDA-70M [57]. Note that we make sure that none of the testing scenes we show below is included in the training datasets.

**Inference cost.** Our model can be flexibly applied to different resolutions and numbers of context views. Measured on a single NVIDIA A100 GPU, BTimer takes $20\,\text{ms}$ for 4-view $256^2$ reconstruction, $150\,\text{ms}$ for the same resolution with 12 views, and $1.55\,\text{s}$ for 12-view $512^2$ reconstruction. It requires less than 10 GB memory, which easily fits on a commercial-grade GPU (Result shown in Supplement). Please note that our model inference can be parallelized and the overall time overhead remains constant given sufficient memory.

## 4.2 Dynamic novel view synthesis

### 4.2.1 Quantitative analysis

We provide quantitative evaluations on two of the largest dynamic view synthesis benchmarks.

**DyCheck benchmark [64].** The benchmark includes a dataset that contains 7 dynamic scenes recorded by 3 synchronized cameras. Following the protocol in [64], we take images from the iPhone camera as our context frames and use the frames from the 2 other stationary cameras for evaluation (totaling 3928 images of resolution $360 \times 480$). Our baselines include per-scene optimization-based methods, *i.e.*, TiNeuVox [63], NSFF [2], T-NeRF [64], Nerfies [32] and HyperNeRF [4]. We additionally compare to a pseudo-feed-forward approach PGDVS [39].

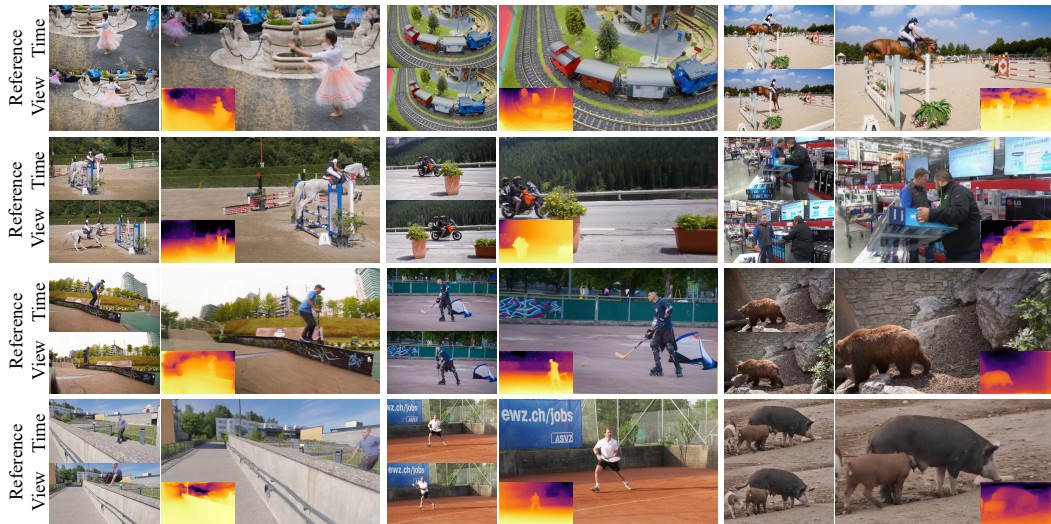

Figure 4: **Visualizations on DAVIS dataset [67].** We show our renderings on novel combinations of view poses and timestamps, with the correspondending references shown on the left. The lower-left/right corner shows the rendered depth map for each example.

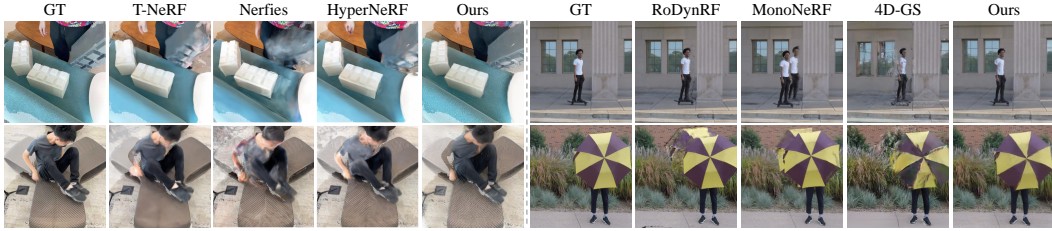

Figure 5: **Qualitative results** on DyCheck [65] (left) and NVIDIA dynamic scene [1] (right) benchmarks.

We report masked Peak Signal-to-Noise Ratio (PSNR), Structural Similarity Index Measure (SSIM) [68], and LPIPS following the benchmark protocol [64] in Tab. 1a, and show visualizations in Fig. 5. Note that since multi-frame inference can run in parallel, for our model we report single-frame reconstruction time regardless of video lengths. It is encouraging to observe that even without per-scene optimization, BTimer achieves a very competitive performance compared to the baselines, ranking 2nd in both SSIM and LPIPS scores. Our model surpasses PGDVS across all 3 metrics without the need of consistent depth estimate. This demonstrates our model's efficiency and strong generalization capability, being capable of providing sharper details and richer textures.

**NVIDIA dynamic scene benchmark [1].** NVIDIA Dynamic Scene dataset contains 9 scenes captured by 12 forward-facing synchronized cameras. Following the protocol in DynNeRF [65], we build the input by selecting the frames at different timestamps in a 'round-robin' manner. Then we evaluate the novel view synthesis quality at the first camera view but at different timestamps. We compare against HyperNeRF [4], DynNeRF [65], RoDynRF [33], 4D-GS [66], Casual-FVS [9] as per-scene optimization baselines.

Our results are shown in Tab. 1b and Fig. 5. Our model demonstrates performance that is competitive or exceeds that of previous optimization-based methods, ranking 3rd among all baselines in terms of PSNR. Compared to the explicit 3DGS-based representation [66, 9], our approach outperforms their performance by 5% on PSNR (25.82dB vs. 24.57dB). In terms of training and rendering speed, NeRF-based methods [65, 6] require multiple GPUs and/or >1 day for optimization. Compared to [66, 9], our feed-forward bullet-time formulation is significantly faster, requiring no optimization time and rendering in real-time.

### 4.2.2 Qualitative analysis

To assess the performance of our method in real-world scenarios, we select multiple monocular videos from the DAVIS dataset [67] for testing. Camera poses for the videos were estimated using

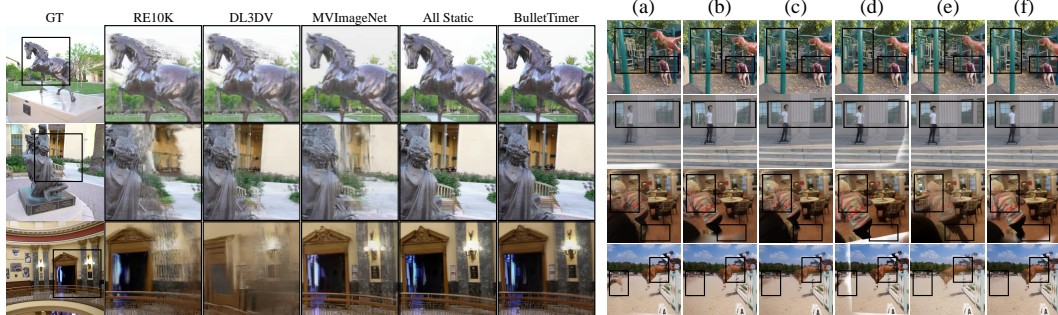

Figure 6: *Left*: **Qualitative comparison of models trained on different datasets** and evaluated on the out-of-distribution Tanks & Temples benchmark [69]. *Right*: (a) model w/o 3D Pretrain, (b) model w/ Re10K only 3D Pretrain, (c) model w/o static Co-train in **Stage 2**, (d) model w/o interpolation supervision, (e) Novel Time Enhancer model, (f) our full model. The upper two scenes are from NVIDIA dataset, and lower two scenes are from DAVIS dataset.

the same annotation technique as detailed in § 3.3. Fig. 4 shows a visualization of the results. Our model demonstrates strong generalization capabilities in real-world captures, producing high-quality, sharp renderings across a variety of objects with complex motions while maintaining robust temporal and multiview consistency.

## 4.3 Compatibility with static scenes

Although our model is primarily designed to handle dynamic scenes, the formulation and the training strategy enable it to be still backward compatible with static scenes. In this section, we show that the *same* model achieves competitive results on static scenes.

**RealEstate10K (RE10K) benchmark [50].** We evaluate our model on the RE10K dataset and compare with several state-of-the-art models [70, 15, 14, 5]. To ensure comparability with baseline models, we train and test our model using 256×256 resolution. Fig. 7a presents a quantitative comparison on LPIPS, where our static model outperforms all the baselines. Please refer to the Supplement for more comparisons on other metrics and visualizations.

| Model | LPIPS↓ |
|---|---|
| GPNR [70] | 0.250 |
| PixelSplat [15] | 0.142 |
| MVSplat [14] | 0.128 |
| GS-LRM [5] | 0.114 |
| **Ours**-Static | 0.070 |
| **Ours**-Full | 0.089 |

(a)

| Model | Datasets | LPIPS↓ |
|---|---|---|
| GS-LRM* [5] | RE10K | 0.310 |
| **Ours**-Static | Objaverse | 0.668 |
| | MVImageNet | 0.343 |
| | DL3DV | 0.278 |
| | All Static | 0.093 |
| **Ours**-Full | +Dynamic | 0.093 |

(b)

Figure 7: **Quantitative comparisons on static datasets.** (a) results on the RE10K benchmark [50]; (b) results on the Tanks and Temples benchmark [69]. We highlight the best, second best, and third best models. *: Our reproduced results.

**Tanks & Temples benchmark [69].** We further evaluate our model on an unseen test dataset, the Tanks & Temples [71] subset from the InstantSplat [69] benchmark, which consists of 10 scenes. We use the state-of-the-art novel view synthesis model [5] as our baseline, reproducing their model since the original code and weights are not publicly available. Additionally, we include our pretrained static model from **Stage 1** as an additional baseline.

To analyze the impact of our mixed-dataset pretraining strategy, we also train single-dataset models using the same training schedule as further baselines. All models utilize 4 context views. Quantitative results (Fig. 7b) demonstrate that our pretrained static model with mixed-dataset training substantially outperforms the single-dataset models, highlighting the crucial role of multi-dataset training for generalization to unseen domains. Even when incorporating the dynamic scene datasets, BTimer achieves comparable result to our best static models. § 4.2.1 provides a qualitative comparison, showing that BTimer consistently generates sharper outputs that closely align with the ground truth.

## 4.4 Ablation study

We study the effect of different design choices. **1) Context frames.** We visualize the reconstruction results as we progressively add 3DGS predictions from more context frames across multiple different

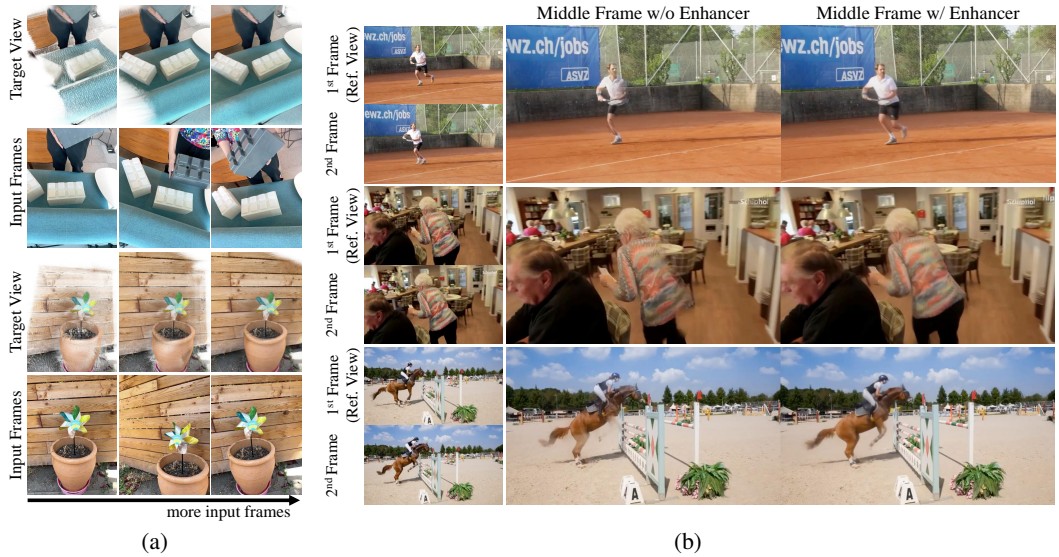

Figure 8: (a) **Illustration of bullet-time reconstruction from multiple context frames.** Increased number of frame predictions leads to progressively more complete scene reconstruction on target views. (b) **Ablation on the NTE module**. The middle frame is in between the 1st frame and the 2nd frame. Results are rendered from the view of the 1st frame.

timestamps in Fig. 8a, where increasing the number of context frame leads to progressively more complete scene reconstruction. This demonstrates the flexibility of our bullet-time reconstruction formulation: during the inference stage, we can arbitrarily select spatially-distant frames that contribute to a more complete view coverage of the scene. **2) Curriculum training.** We show in § 4.2.1 the effect of our curriculum training strategy. Without **Stage 1** of pre-training on static scenes, the model struggles to produce results of correct geometry and sharp details. Pretraining on multiple diverse datasets is also crucial, which we demonstrate by *just* training on RE10K dataset, and non-negligible distortions are observed in the results. Similarly, even in **Stage 2** of our curriculum, we still need to co-train on static scenes which provide more multi-view supervisions, thus maintaining the rich details and reasonable geometries. Quantitative ablation results are shown in the Supplement. **3) Interpolation supervision.** Shown in § 4.2.1 (with more results in the Supplement), interpolation supervision (introduced in § 3.1) plays a significant role, without which the model tends to produce *white-edge* artifacts. This occurs because without interpolation loss, the model often generates 3DGS that are positioned too close to the camera with low depth values to *cheat* the loss. In contrast, adding the interpolation supervision requires the model to account for scene dynamics and encourages consistency across multiple views. **4) NTE.** As demonstrated in Fig. 8b, our NTE module enhances the bullet-time reconstruction model's ability to handle scenes with fast or complex motions, largely reducing the ghosting artifacts. Additional video results are provided in the supplementary material. Although 3D-free design enables NTE to handle complex dynamics and produce smooth transitions between adjacent frames, the model struggles to produce novel views that are far from the input camera trajectory (As illustrated in § 4.2.1).

## 5 Conclusion

In this paper we present BTimer, the first feed-forward dynamic 3D scene reconstruction model for novel view synthesis. We present a bullet-time formulation that allows us to train the model in a more flexible and scalable way. We demonstrate through extensive experiments that our model is able to provide high-quality results at arbitrary novel views and timestamps, outperforming the baselines in terms of both quality and efficiency.

**Limitations.** Our method is mainly targeted for novel view synthesis, and the recovered geometry (hence the depth map) is usually not as accurate. Correspondences between frames are implicitly modeled by the neural network, and our pixel-aligned Gaussian representation cannot represent temporal deformations. Although practically we observe temporally coherent results, additional post-processing steps have to be introduced to recover the explicit motion of the geometry.

**Broader Impact.** BTimer can transform posed casual videos into realistic dynamic 3D assets. However, it should be used with caution, particularly concerning privacy, copyrights, and the potential for malicious impersonation.

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

# Supplementary Material

In this supplementary material, we provide additional details on the datasets used in our experiments (§ F) and our training cost and more ablation studies (§ G). On the results side, we first show more qualitative static and dynamic reconstruction results (§ H, § I). We further justify our choice of the bullet-time formulation by showing that our reconstruction is **temporally smooth** (§ J), as well as an simple extension to unlock the power for **estimating dynamic deformations** (§ K).

## F    Dataset Details

The static datasets used in our training are as follows: OBJAVERSE [49] is a synthetic object-centric dataset, and we use the 80K-object subset from [72]. MVIMGNET [51] is a real-world object-centric dataset that has 220K objects. RE10K [50] is a real-world scene dataset that has 80K video clips. DL3DV [52] is a real-world scene dataset that has 10K video. We sample DL3DV 10 times more frequently than other datasets to balance the number of training samples. We use a spatial scale of 8 for Objaverse and scale 1 for all other datasets.

The dynamic datasets used in our training are as follows: KUBRICMV is a synthetic multi-vew video dataset that has 3K scenes. We rendered this dataset using the Kubric [53] engine. The scene setup follows Movi-E [53] and videos are rendered from all camera poses in the camera trajectory so it produces a multi-view video. POINTODYSSEY [54] is a synthetic monocular dataset with 131 scenes. DYNAMICREPLICA [55] is a synthetic stereo video dataset with 484 training sequences. SPRING [56] is a synthetic stereo video dataset with 37 scenes. PANDA-70M [57] is a real-world monocular video dataset. We use around 40K clips filtered from a random subset. We use scale 6 for Spring and DynamicReplica and 1 for other datasets. More details can be found in Tab. S2

| Dataset | Dynamic | Subject | Domain | #Views | #Frames | #Scenes | #Multiplies | #Scale |
|---|---|---|---|---|---|---|---|---|
| RE10K [50] | | *S* | Real | - | 10M | 80K | 1 | 1 |
| MVImgNet [51] | | *O* | Real | - | 6.5M | 220K | 1 | 1 |
| Objaverse [49] | | *O* | Synthetic | - | 4M | 80K | 1 | 8 |
| DL3DV [52] | | *S* | Real | - | 51M | 10K | 10 | 1 |
| PointOdyssey [54] | ✓ | *O+S* | Synthetic | 1 | 6K | 131 | 3e3 | 1 |
| Kubric-MV [53] | ✓ | *O+S* | Synthetic | 24 | 70K | 3K | 2e2 | 1 |
| DynamicReplica [55] | ✓ | *O+S* | Synthetic | 2 | 145K | 484 | 8e2 | 6 |
| Spring [56] | ✓ | *O+S* | Synthetic | 2 | 200K | 37 | 1e4 | 6 |
| PANDA-70M [57] | ✓ | *O+S* | Real | 1 | 19M | 40K | 10 | 1 |

Table S2: **Datasets. Dynamic** indicates if the dataset is dynamic or static. **Subject** indicates if the dataset is object-centric (*O*) or scene-centric (*S*). **Domain** indicates if the dataset is captured from the real world or is synthesized. **#Views** denotes the number of synchronized views for a dynamic video. **#Frames** and **#Scenes** are the numbers of image frames and unique scenes in the dataset respectively. **#Multiplies** denotes the number of multiplies we sample the dataset (by scene) in training for balance. **#Scale** is the scale we applied to the dataset so that all datasets have approximately the same metric scale.

## G    Training Cost Analysis and Effect of Batch Size

The full training of BulletTimer takes ~4 days on 32 NVIDIA A100 GPUs. As illustrated in Fig. S9, the training cost is comparable to existing feed-forward 3D reconstruction methods, such as LVSM [16] and LRM [12] (384 GPU-days) or GS-LRM [5] (192 GPU-days). Like these methods, our work also functions as an amortized algorithm: once trained, the inference cost becomes negligible. Taking inference cost also into consideration, per-scene optimization quickly becomes more expensive, with the difference becoming more pronounced with the growing number of scenes.

Fig. S10 shows the results of training our model with 1 GPU, 8 GPUs, and 32 GPUs. Although inference fits on a single GPU, our training benefits from large batch sizes so we used 32 GPUs (each

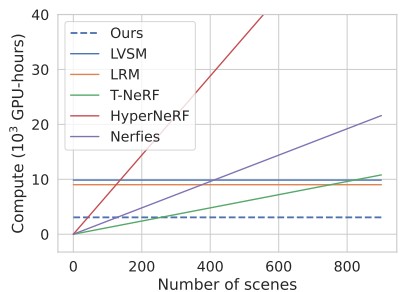

Figure S9: Computation cost.

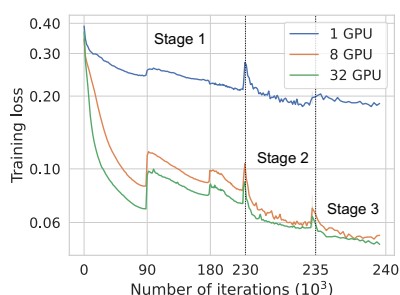

Figure S10: Batch-size ablation.

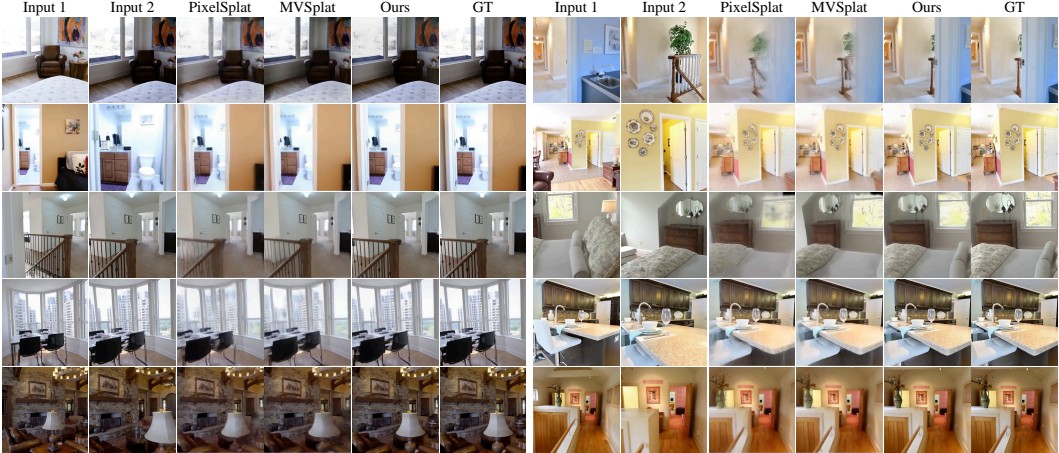

Figure S11: **Comparison on static scene dataset.** We compare our renderings with the baseline models, trained and tested on the RE10K dataset.

GPU holds a single batch). The same number of GPUs was also used in both LVSM and GS-LRM. In line with other fields (LLMs, GenAI), we regard the scalability of our method one of its key strengths. For ease of reproduction and fine-tuning, we will release our source code and pretrained checkpoints. To provide a more rigorous assessment of runtime, we have conducted an additional study varying both the number of context frames and the input resolution (see Tab. S8). The results show that BTimer consistently remains in the second range (even at 512×512 resolution with 12 context frames), and is still several orders of magnitude faster than per-scene optimisation pipelines.

## H  More Results on Dynamic Scenes

### H.1  Qualitative Results

We have provided video results on the DyCheck Benchmark [65] and NVIDIA Dynamic Scene Benchmark [2] on our project webpage `https://research.nvidia.com/labs/toronto-ai/bullet-timer/`. Additionally, we include novel view synthesis videos for the DAVIS dataset, DyCheck iPhone dataset, and SORA scenes. We also showcase a video demonstrating the effects of the NTE module, along with our video results on the Tanks & Temples static scenes.

### H.2  Additional Quantitative Results

In this section, we compare our method with more sota models on DyCheck iPhone dataset [64] (see Tab. S5a) and NVIDIA Dynamic Scene dataset [1] (see S5b). All previous models require hours of per-scene optimization, whereas our method performs inference in real time with comparable accuracy. We also compare with DynIBaR[8] using DynIBaR's protocol, which demonstrates strong performance on the NVIDIA Dynamic Scene Benchmark(see Tab. S4). While our approach performs

| Model | PSNR↑ | SSIM↑ | LPIPS↓ |
|---|---|---|---|
| PixelNeRF [73] | 20.43 | 0.589 | 0.550 |
| GPNR [70] | 24.11 | 0.793 | 0.250 |
| AttnRend [74] | 24.78 | 0.820 | 0.213 |
| MuRF [75] | 26.10 | 0.858 | 0.143 |
| PixelSplat [15] | 25.89 | 0.858 | 0.142 |
| MVSplat [14] | 26.39 | 0.869 | 0.128 |
| GS-LRM [5] | 28.10 | 0.892 | 0.114 |
| **Ours**-Static | 26.49 | 0.886 | 0.096 |
| **Ours**-Static$^\dagger$ | 28.91 | 0.920 | 0.070 |
| **Ours**-BTimer | 26.82 | 0.891 | 0.089 |

Table S3: **Quantitative comparison of models performance on RE10K test set.** To be consistent with the baselines, we adopt the $256 \times 256$ resolution. Our Bullet Timer has been trained on both static and dynamic scenes, while the other model is only trained on RE10K training set. We highlight the best , second best and third best models. †: 4 input views.

slightly worse in terms of reconstruction quality, DynIBaR relies on extensive per-scene optimization (approximately 300 hours per NSFF sequence) and leverages additional supervision signals such as optical flow and depth. In contrast, BTimer is a fully feed-forward system: it reconstructs a scene in just one second and is trained solely with a photometric loss.

| Model | PSNR↑ | SSIM↑ | LPIPS↓ |
|---|---|---|---|
| DynIBaR [8] | 30.86 | 0.957 | 0.027 |
| **Ours**-BTimer | 26.31 | 0.852 | 0.0730 |

Table S4: Quantitative comparison on NSFF benchmark using DynIBaR's protocol

# I More Results on Static Datasets

We provide a comprehensive qualitative comparison of our method against the baselines, MVS-plat [14] and PixelSplat [15], on the RE10K dataset, as shown in Fig. S11. For each scene, the figure also displays the input views provided to the networks. Compared to the baselines, our method produces sharper outputs and more closely aligns with the ground-truth renderings. Note that all the methods used for the evaluation in this figure are trained exclusively on RE10K. Additionally, we use two views as context for all methods to ensure fairness in evaluation and to align with the setup of the baselines. Tab. S3 presents a quantitative evaluation against the baselines under the same settings. While our static model achieves the best performance among the baselines, our dynamic *BTimer* model, trained for the dynamic task, also demonstrates strong performance on the static task, ranking third on the static benchmark.

Our complete static model, trained across all datasets, is capable of reconstructing a highly diverse set of environments. Fig. S12 showcases our model's reconstructions across a wide variety of scenes, including outdoor forward-facing, outdoor drone shots, outdoor 360-degree views, indoor 360-degree views, and indoor forward-facing scenes, as well as object-centric synthetic scenes. Notably, all these reconstructions are achieved using a shared set of weights, demonstrating that our model, trained across multiple datasets, generalizes effectively to different scenarios.

To further demonstrate the importance of training on multiple datasets for generalization to unseen datasets, we conduct an ablation study on the datasets used to train our static model. Tab. S6 compares the performance of our model when trained individually on a single dataset—RE10K, MVImageNet, DL3DV, or Objaverse—against its performance when trained on all these datasets simultaneously. The evaluation is conducted on a completely unseen dataset, the Tanks & Temples split from the InstantSplat [69] paper. Our model, whether static or dynamic, trained on all datasets significantly outperforms the single-dataset models.

| Model | PSNR↑ | SSIM↑ | LPIPS↓ |
|---|---|---|---|
| 4D GS [66] | 13.64 | - | 0.428 |
| Gauss.Marbles [76] | 16.72 | - | 0.413 |
| DyBluRF [77] | 17.37 | 0.591 | 0.373 |
| D-NPC [78] | 16.41 | 0.582 | 0.319 |
| Shape-of-Motion [10] | 17.32 | 0.598 | 0.296 |
| MoSca [79] | 19.32 | 0.706 | 0.264 |
| PGDVS [39] | 15.88 | 0.548 | 0.340 |
| Depth Warp | 7.81 | 0.201 | 0.678 |
| BTimer (**Ours**) | 16.52 | 0.570 | 0.338 |

(a)

| Model | PSNR↑ | LPIPS↓ |
|---|---|---|
| D-NeRF [30] | 21.49 | 0.232 |
| NR-NeRF [80] | 19.69 | 0.323 |
| TiNeuVox [63] | 19.74 | 0.285 |
| NSFF [2] | 24.33 | 0.199 |
| MonoNeRF [6] | 25.62 | 0.106 |
| DynPoint [81] | 26.53 | 0.068 |
| D-NPC [78] | 25.64 | 0.109 |
| MoSca [11] | 26.72 | 0.070 |
| PGDVS [39] | 24.41 | 0.186 |
| Depth Warp | 12.63 | 0.564 |
| BTimer (**Ours**) | 25.82 | 0.086 |

(b)

Table S5: **Additional quantitative comparisons on dynamic datasets.** (a) DyCheck iPhone dataset [64]. (b) NVIDIA Dynamic Scene dataset [1].

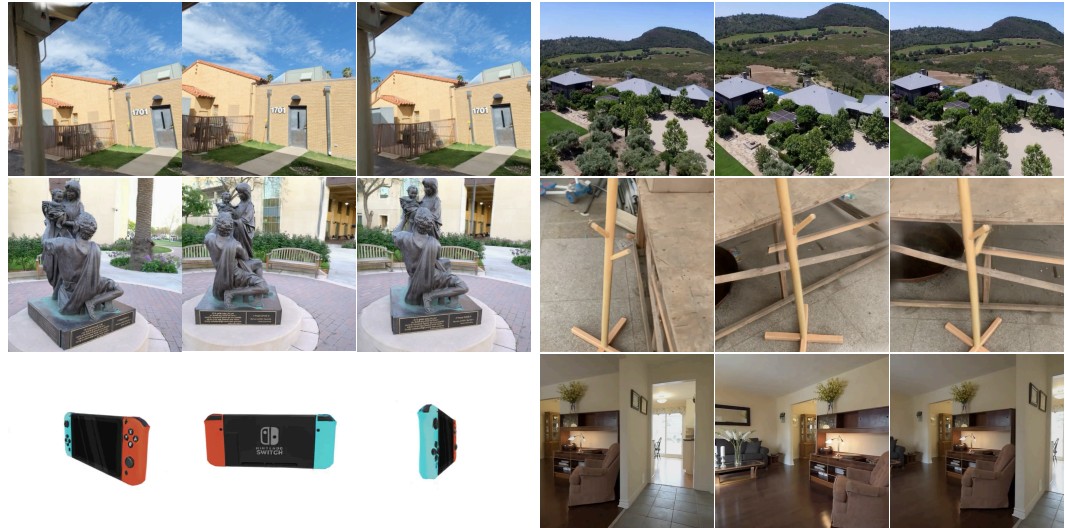

Figure S12: A diverse set of scenes reconstructed using our static model, trained on multiple datasets and capable of generalizing to various scenarios.

## J   Evaluation of Temporal Smoothness

Since our method reconstructs each timestamp individually, it is necessary to understand its temporal smoothness. In this section, we quantitatively evaluate the temporal smoothness of the reconstructed dynamic scenes, with results shown in Tab. S9. We render the reconstructed DyCheck [65] scenes from one of the evaluation fixed cameras and evaluate the rendered video using the *Temporal Flickering* metric in VBench [83]. Concretely, the metric computes the pixel-wise Mean Absolute Error in every two adjacent frames and averages over all pixels and frames:

$$S_{\text{flicker}} = \frac{1}{N} \sum_{i=1}^{N} \left( \frac{1}{T-1} \sum_{t=1}^{T-1} MAE(f_i^t, f_i^{t+1}) \right), \tag{S.2}$$

where $N$ is the number of videos, $T$ is the number of frames per video, $f_i^t$ is the frame $t$ in video $i$, and $MAE$ is the Mean Absolute Error between two consecutive frames over all pixel locations. Finally, the metric is normalized into the range of 0 to 1:

$$S_{\text{flicker-norm}} = \frac{255 - S_{flicker}}{255}. \tag{S.3}$$

The higher the metric, the less flickering will be observed in a video. BTimer achieves the second best on the Temporal Flickering metric, which suggests that our bullet-time prediction, though not explicitly associated across frames, still achieves a better temporal smoothness than other baselines that decode from some temporal representations.

| Model | Datasets | PSNR↑ | SSIM↑ | LPIPS↓ |
|---|---|---|---|---|
| GS-LRM* [5] | RE10K | 17.56 | 0.546 | 0.310 |
| **Ours**-Static | Objaverse | 7.00 | 0.363 | 0.668 |
| | MVImageNet | 17.75 | 0.530 | 0.343 |
| | DL3DV | 17.92 | 0.566 | 0.278 |
| | All Static | 24.22 | 0.807 | 0.093 |
| **Ours**-Full | +Dynamic | 24.13 | 0.806 | 0.093 |

Table S6: **Baseline comparisons on the Tanks & Temples dataset (InstantSplat split).** Test views are $512 \times 512$. LPIPS are compuated on $256 \times 256$. We highlight the best , second best and third best models. *: Our reproduced results.

| Method | PSNR ↑ |
|---|---|
| w/o 3D Pretrain | 17.94 |
| w/ Re10K only 3D Pretrain | 21.29 |
| w/o static Co-train | 22.79 |
| w/o interpolation supervision | 20.54 |
| Full model | **24.00** |

Table S7: Quantitative ablation results on NVIDIA Dynamic Scene Benchmark. Ablation models are trained with 4 context frames.

| #Ctx. | Res. | Time | Mem. |
|---|---|---|---|
| 4 | $256^2$ | 0.02s | 1.42G |
| 12 | $256^2$ | 0.15s | 2.60G |
| 12 | $512^2$ | 1.55s | 9.68G |

Table S8: Inference cost. Model is evaluated on a single NVIDIA A100 GPU.

# K   Visualization of Learned Deformation

While BTimer is primarily intended for novel view synthesis at the bullet timestamp, in order to demonstrate that our model design can be also targeted for building explicit temporal correlations, we train a variant of BTimer that predicts the canonical positions (XYZ) of Gaussians instead of the pixel-aligned depths on the Objaverse4D [17] dataset. The 4 input images are taken from different camera poses and different timestamps. In Fig. S13, we render the reconstructed dynamic object from a fixed viewpoint and find that the model successfully recovers the 3D motion by predicting the positions of the Gaussians at the correctly warped locations. This is further justified by keeping only the partial reconstruction from the 3DGS associated with one of the input images, and we find out that the model learns to warp the Gaussians according to different timesteps. Canonical XYZ prediction is commonly used in object-centric cases for being bounded (LGM [72], L4GM [17]), however pixel-aligned prediction is popular in large unbounded scene for being easy to optimize (pixelSplat [15], GS-LRM [5]). Applying canonical XYZ prediction to scene data is a valuable direction that we would like to explore in future work.

# L   Visualization of Learned Scene Flow

Although BTimer is not trained with scene flow supervision, we show that our model effectively model scene flows under the hood in the process of learning dynamic reconstruction. We treat the Gaussian associated with each pixel in the input images as a point and treat its trajectory over time as a scene flow. The visualization in Fig. S14 suggests that the learned scene flows closely represent the actual object motion.

| Model | apple | block | windmill | space | spin | teddy | wheel | Average |
|---|---|---|---|---|---|---|---|---|
| Ground Truth | 0.9878 | 0.9767 | 0.9940 | 0.9939 | 0.9829 | 0.9759 | 0.9650 | 0.9823 |
| TiNeuVox [63] | 0.9807 | 0.9814 | 0.9879 | 0.9949 | 0.9832 | 0.9782 | 0.9695 | 0.9823 |
| T-NeRF [64] | 0.9831 | 0.9791 | 0.9866 | 0.9907 | 0.9828 | 0.9730 | 0.9624 | 0.9797 |
| Nerfies [32] | 0.9817 | 0.9791 | 0.9868 | 0.9918 | 0.9809 | 0.9720 | 0.9609 | 0.9790 |
| HyperNeRF [4] | 0.9825 | 0.9784 | 0.9865 | 0.9914 | 0.9821 | 0.9720 | 0.9584 | 0.9787 |
| PGDVS [39] | 0.9719 | 0.9738 | 0.9956 | 0.9903 | 0.9816 | 0.9649 | 0.9517 | 0.9757 |
| BTimer (**Ours**) | 0.9835 | 0.9760 | 0.9884 | 0.9881 | 0.9789 | 0.9746 | 0.9745 | 0.9806 |

Table S9: **Temporal Flickering [82] evaluation on the DyCheck [65] dataset.**. There are 7 scenes and we report their average. We highlight the best , second best and third best .

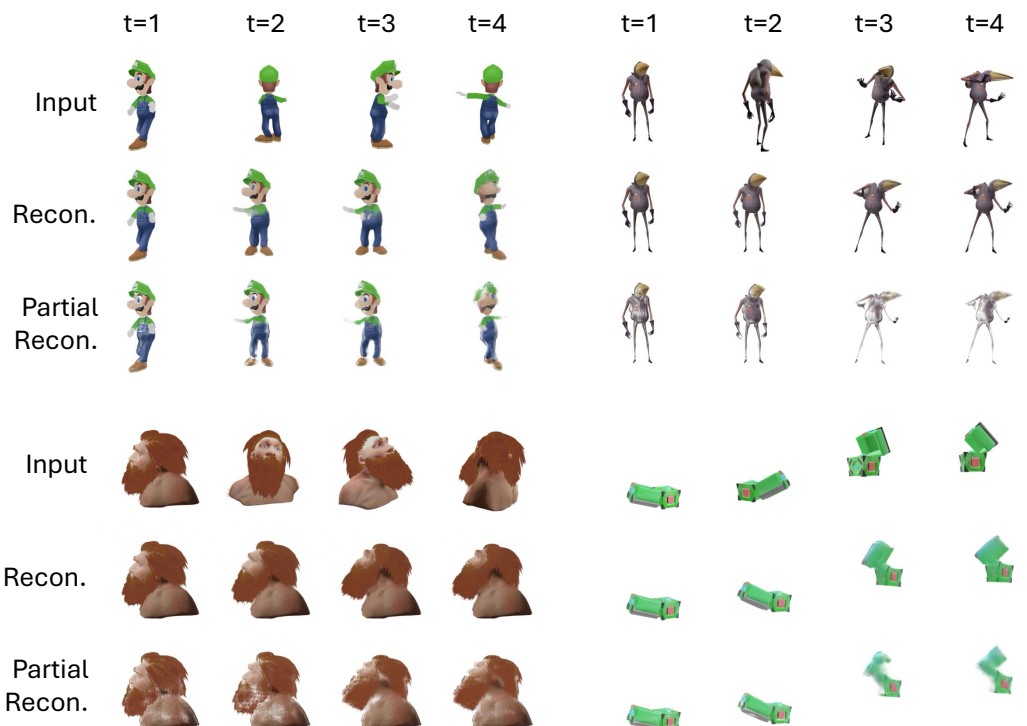

Figure S13: **Learned deformation visualization.** In each example, the first row shows the 4 input images captured from different viewpoints and timestamps, the second row is our reconstruction rendered from a fixed viewpoint, and the third row keeps only Gaussians associated from one of the input images.

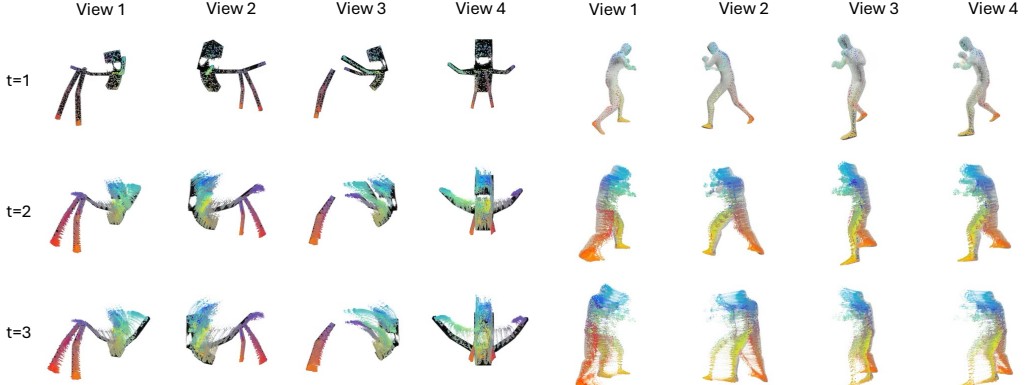

Figure S14: **Learned scene flow visualization.** We color the Gaussians by the pixel positions they associate with. As the Gaussians move, their trajectories are considered as scene flows. Our model learns meaningful scene flows that closely represent the object motion without any scene flow supervision.

