# OpenReview forum: "Feed-Forward Bullet-Time Reconstruction of Dynamic Scenes from Monocular Videos"
_NeurIPS.cc/2025/Conference — NeurIPS 2025 poster_

### Official Review · Reviewer_69bn · 2025-06-30

**Clarity:** 3
**Significance:** 3
**Originality:** 3
**Rating:** 4
**Confidence:** 3

**Summary:**

This paper presents BTimer, a novel feed-forward model for real-time 3D scene reconstruction and novel view synthesis from monocular videos. Its key innovation is the introduction of a bullet-time formulation that enables flexible, scalable training and inference across arbitrary timestamps. BTimer leverages a neural rendering approach using 3D Gaussian splatting, which allows it to efficiently generate high-quality, temporally coherent views without the need for per-scene optimization. The model can handle dynamic scenes effectively, producing detailed reconstructions and novel views with minimal latency—around 150ms—outperforming prior optimization-based methods in speed, quality, and generalization. The work also demonstrates compatibility with static scenes and discusses potential applications and broader impacts.

**Questions:**

The pipeline for annotating internet videos—using SAM to mask dynamic objects before applying DROID-SLAM—is a very clever approach to leverage a powerful static-scene SLAM system. However, this relies on the segmentation quality and treats motion as an outlier to be removed, rather than data to be modeled.

I'm curious if the authors experimented with SLAM systems that explicitly model dynamic objects? For example, methods that track moving objects separately or represent dynamic parts of the scene with their own motion models. What was the rationale for choosing the current pipeline over these alternatives? Was it primarily for robustness and scalability on diverse internet videos, or did dynamic-aware SLAM methods prove less reliable in practice?

**Ethical Concerns:**

["NO or VERY MINOR ethics concerns only"]

**Limitations:**

Gaps in Ablation and Analysis：\
While the paper's ablation studies are generally solid, certain aspects could benefit from further exploration. For instance, using DROID-SLAM and SAM to annotate web videos is a crucial part of the data strategy. However, the robustness to annotation errors and the impact of these potential errors on final performance have not been quantitatively analyzed.

**Paper Formatting Concerns:**

No issues.

**Quality:**

3

**Strengths And Weaknesses:**

Strengths:
1. Novelty and Significance: It proposes BTimer, the first feed-forward model for this task, using an elegant "bullet-time" formulation to unify static and dynamic scene reconstruction.
2. Performance: It achieves a massive speedup (reconstruction in under a second versus hours for competitors) while maintaining competitive, state-of-the-art quality on major benchmarks.
3. High Quality: The technical approach is sound, using a ViT and 3DGS backbone, and is validated by comprehensive experiments and ablation studies that demonstrate the effectiveness of its curriculum training and Novel Time Enhancer (NTE) module.
4. Clarity and Presentation: The paper is well-written and easy to follow. Its core ideas are supported by high-quality visualizations and thorough experimental results that convincingly validate its claims.

Weakness:
1. Dependencies: The model relies on pre-computed camera poses and requires significant computational resources for the initial training.
2. Implicit Motion: Dynamics are modeled implicitly, without explicit motion fields, which limits applicability for editing or physical analysis.

---

> ### Author Rebuttal · Authors · 2025-07-30
>
> > **[W1]** Dependencies on pre-computed camera poses and requires significant computation for the initial training.
>
> 1. Most feed-forward reconstruction models need camera poses as input. Particularly in monocular dynamic scene reconstruction settings, without camera poses will add more ambiguity to the dynamics of the scene, so we use it in this work. And we would like to leave the model without camera poses for future work. We will also release the full method and code for performing camera annotations for monocular video soon.
> 2. The training cost for BTimer (≈ 256 GPU‑days) is on par with other state‑of‑the‑art feed‑forward 3 D reconstruction methods—e.g., LVSM (ICLR’25) (≈ 384 GPU‑days), LRM (ICLR’24) (≈ 384 GPU‑days), and GS‑LRM (ECCV’24) (≈ 192 GPU‑days). Like these works, our model operates as an amortised algorithm: once training is complete, inference is virtually negligible. When inference cost is taken into account, per‑scene optimization pipelines rapidly become more expensive, and the gap widens as the number of scenes grows (see Fig. S1 in the supplementary).
>
>
>
> > **[W2]** Implicit Motion
>
> 1. Modeling dynamics implicitly allows our model to capture flexible and complex dynamics, such as deformation with topological changes, and complex natural phenomena such as fire, waves, smoke, and dust. This is evident in our supplementary visualizations on SORA and DAVIS datasets.
> 2. Despite using implicit time encoding, our model can still recover decent 3D deformation and scene flows as evidenced in Figure S5 and S6 of the supplementary, where the dynamic structure of the scenes is clearly recovered. This indicates that the model learns meaningful motion representations from time embeddings, even without explicit motion supervision.
> 3. Our approach aligns with several prior works that successfully use implicit time modeling for dynamic scenes, including ST-NeRF (CVPR’21), DyNeRF (CVPR’22), and HexPlane (CVPR’23).
>
>
>
> > **[Q1]** The pipeline for annotating internet videos—using SAM to mask dynamic objects before applying DROID-SLAM—is a very clever approach to leverage a powerful static-scene SLAM system. However, this relies on the segmentation quality and treats motion as an outlier to be removed, rather than data to be modeled.
>
> Thanks for the thoughtful comment. We generally observe that the segmentation quality from SAM is sufficient for the majority of internet videos. In challenging cases where semantic segmentation may fail, the original outlier rejection mechanism in DROID‑SLAM provides a complementary safeguard by regressing per‑pixel weights, which we use as a supplement to down‑weight and prevent them from corrupting the pose estimation.
>
> It is important to note that, in this pipeline, our primary objective is to obtain reliable camera poses. As such, explicit motion modeling at this stage is not strictly required, which is why we remove them rather than model them directly. Motion information is preserved in the input frames and later learned by the BTimer network during reconstruction.
>
>
>
> > **[Q2]** I'm curious if the authors experimented with SLAM systems that explicitly model dynamic objects? For example, methods that track moving objects separately or represent dynamic parts of the scene with their own motion models. What was the rationale for choosing the current pipeline over these alternatives? Was it primarily for robustness and scalability on diverse internet videos, or did dynamic-aware SLAM methods prove less reliable in practice?
>
> Thank you for the question. We did consider SLAM‑style pipelines that treat dynamic objects separately, but found them ill‑suited to our target setting of unconstrained, monocular in-the-wild videos:
> 1. Limited robustness and scalability. Dynamic‑aware SLAM pipelines require reliable depth, motion segmentation, and object tracking. In practice these steps are not robust enough (PGDVS) and fail on texture‑poor, fast‑moving or out-of-distribution clips. Per‑object tracking and optimisation make such systems slower than a single feed‑forward pass and difficult to scale to large datasets.
> 2. And SLAM methods typically assume known and accurate camera intrinsics, which are unavailable for the vast majority of in‑the‑wild videos.
> With the goal of pursuing robustness, scalability and a complete generalizable framework, we make this framework design.
>
>
>
> > **[Limitation1]** Gaps  in Ablation and Analysis
>
> Thank you for the suggestion. Although we have not yet performed a dedicated robustness study, several design choices (please refer to answer of Question1) already mitigate most SAM + SLAM annotation errors. We will add a detailed discussion of these factors in the final version.

---

> > ### Author Response · Authors · 2025-08-04
> >
> > Dear reviewer `69bn`,
> >
> > We would be grateful if you could let us know whether our rebuttal has addressed your concerns. Please don’t hesitate to reach out if you have any remaining questions or suggestions. We sincerely appreciate the time and effort you have dedicated to reviewing our paper.

---

> > > ### Comment · Reviewer_69bn · 2025-08-04
> > >
> > > Thank you for the detailed rebuttal and for your comprehensive responses to my questions.
> > >
> > > To follow up on our discussion, here are some questions I have for you:
> > >
> > > 1. Could you please explain the key differences between implicit and explicit approaches in this context, especially concerning how they affect model design and training?
> > >
> > > 2. Given that the performance gains appear to be incremental, could you elaborate on the specific scenarios or applications where these improvements would be most impactful and justify the added complexity?

---

> > > > ### Author Response · Authors · 2025-08-04
> > > >
> > > > **[Q1]** Could you please explain the key differences between implicit and explicit approaches in this context, especially concerning how they affect model design and training?
> > > >
> > > > 1. In our context, explicit approaches refer to methods that directly model scene dynamics, typically through mechanisms like scene flow or velocity, which represent how 3D points move over time. From a model design perspective, some works adopt a canonical field combined with a deformation field (e.g., D-NeRF [CVPR’21], 4DGS [CVPR’24]), while others define motions explicitly using velocity vectors (e.g., STORM [ICLR’25]). These designs are common in per-scene optimization-based dynamic reconstruction, where the dynamics are encoded and optimized for each scene individually. Such models often rely on additional supervision signals (e.g., depth, optical flow) to ensure accurate estimation of motion fields.
> > > >
> > > >    (Note: Although ST-NeRF [CVPR’21] uses time encoding, it actually models an explicit deformation field and thus falls under the explicit category.)
> > > >
> > > > 2. In contrast, implicit approaches, like ours, do not explicitly model the dynamics. Instead, we introduce a time embedding into the model, which implicitly encodes temporal information. The network learns the scene dynamics directly from data without relying on explicitly defined deformation or motion fields. This design is similar to works like DyNeRF [CVPR’22] and HexPlane [CVPR’23].
> > > >
> > > >    Empirically, we find that the implicit design is easier to train in a feed-forward setting and more robust in large-scale training, as it avoids the complexities of explicit scene flow estimation and does not require additional supervision. Moreover, it enables our model to be trained in a unified manner across both static and dynamic scenes, offering better scalability and generalization.
> > > >
> > > > We hope this clarifies the distinction.
> > > >
> > > > **[Q2]** Given that the performance gains appear to be incremental, could you elaborate on the specific scenarios or applications where these improvements would be most impactful and justify the added complexity?
> > > > Our design offers improved robustness, flexibility, and scalability in practical settings:
> > > > 1. We experimentally find that while methods that decode deformation fields for each 3D Gaussian (e.g., 4DGS [CVPR’24] built on HexPlane) can successfully overfit individual 4D sequences, they struggle to learn stable deformation fields in large-scale, feed-forward training settings, as also observed in L4GM [NeurIPS’24]. In contrast, our implicit modeling approach leads to more stable training and generalization well across scenes.
> > > > 2. Models like STORM [ICLR’25], which assume linear motion, are well-suited for AV data but lack flexibility for complex, non-rigid dynamics. Our approach generalizes to a wider range of scenes, including challenging cases with unstructured motion.
> > > > 3. Though implicit, our model still recovers decent 3D deformations and flows (see Figs. S5/S6 of the supplementary), enabling applications such as editing or physical analysis.
> > > > 4. By modeling dynamics implicitly, our approach can capture rich, highly non-linear temporal behaviors, such as fire, water, smoke, and dust, which are difficult to represent using explicit or linear motion models. This is evident in the DAVIS and SORA scenes shown in our supplementary materials.

---

> > > > > ### Comment · Reviewer_69bn · 2025-08-05
> > > > >
> > > > > Thanks for your detailed responses. Most of my concerns have been addressed, and I will maintain my original score of 4.

---

### Official Review · Reviewer_6kVt · 2025-07-01

**Clarity:** 3
**Significance:** 4
**Originality:** 3
**Rating:** 5
**Confidence:** 4

**Summary:**

BTimer is a motion-aware, feed-forward model designed for real-time reconstruction and novel view synthesis of dynamic scenes. Its architecture is both simple and efficient for representing 4D dynamic scenes. The method achieves state-of-the-art performance on both static and dynamic scene datasets, including the DyCheck iPhone, NVIDIA Dynamic Scene, RE10K, and Tanks and Temples benchmarks, proving competitive even against optimization-based approaches.

**Questions:**

1. For a given in-the-wild video, what is the prescribed method for estimating camera poses and intrinsics?

**2. Do the authors intend to release the source code and model checkpoints for re-implementation?**

**Ethical Concerns:**

["NO or VERY MINOR ethics concerns only"]

**Final Justification:**

Most of my concerns have been addressed, so I keep my positive score.
I hope the authors keep their promise and incorporate the discussions and experiments (including the conclusions from other reviews) into the revision.

**Limitations:**

Yes, the paper appropriately discusses its limitations in Section 5.

**Paper Formatting Concerns:**

The paper adheres to the NeurIPS formatting guidelines and is well-organized.

**Quality:**

3

**Strengths And Weaknesses:**

**Strengths:**

1. The paper is well-written, presenting a clear methodology and a logically sound model architecture.

2. The proposed model demonstrates robust performance across a diverse array of datasets, including both static and dynamic scenes. Its effectiveness is validated on real-world captures, AI-generated videos from models like Sora, and rendered scenes from the Objaverse4D dataset.

3. The work addresses a significant and challenging problem within the field, and the efficacy of the proposed method is substantiated through extensive experimentation.

**Weaknesses:**

**0. Do the authors intend to release the source code and model checkpoints for re-implementation? It will be very helpful for the 4D reconstruction of the community.**

1. To prevent "cherry-picking of results", it is recommended that the evaluation metrics and baselines remain consistent throughout the paper. For instance, all metrics, including Rec. Time, Render FPS, PSNR↑, SSIM↑, and LPIPS↓, should be comprehensively reported for all methods in Table 1 (a) and (b).

2. The authors claim a reconstruction time of BTimer of less than 150ms (line 11). However, as indicated in **Table 1** and **Table S5**, this duration is dependent on both the resolution and the number of frames in the input video. While the efficiency of the BTimer's feed-forward architecture is acknowledged, a more qualified and rigorous conclusion regarding its runtime performance is warranted.

3. Given the importance of the deformation field and scene flow for downstream applications, a discussion comparing pixel-aligned models with BTimer's approach of predicting canonical Gaussian positions (XYZ) would be a valuable addition to the main paper.


Minors:

The item labeled as "Figure 7"  is a "table" and appears to be incorrectly captioned.

---

> ### Author Rebuttal · Authors · 2025-07-30
>
> > **[W1]** Benchmark Comparison
>
> Thank you so much for this suggestion.
> 1. Baselines selection: we kindly clarify that we are not cherry-picking baselines. For each benchmark, we compare with methods that have official results on that benchmark, to ensure the comparison is rigorous and fair. This is exactly the same convention followed by previous dynamic‑scene reconstruction works (e.g., MoSca, CAT4D, PGDVS, Casual‑FVS, RoDynRF), where they likewise report different baseline sets across DyCheck and NVIDIA baselines for the same reason. If additional official results appear, we are happy to include these additional comparisons in the revised version.
> 2. Metric Comparison: For the same reason as baselines selections, to enable direct, fair comparisons with the published results of prior work, we follow each benchmark’s established protocol and adopt exactly the metrics that the existing baselines report.  On NVIDIA Dynamic Scenes, prior works (MoSca, Casual‑FVS, RoDynRF, MonoNeRF, DynNeRF) routinely list PSNR, LPIPS but not SSIM. We are happy to provide our SSIM score (`0.835`), however we didn’t find any official reported SSIM score from our compared baselines. FPS is reported in Casual‑FVS, so we adopt the same set. The reconstruction time for both two benchmarks have been reported in PGDVS.
>
>     In any case, we will include complete metric comparison for those that are officially available in our revised version. As for now, we additionally evaluate our methods all three different evaluation protocols used by prior works on Dynamic Scenes, so that Reviewers can conveniently compare.
>
>     || Reference |PSNR↑ | SSIM↑ | LPIPS↓|
>     |--|--|--|--|--|
>     |Protocol A*|  DynNeRF [1] | 25.82 | 0.835 | 0.086 |
>     |Protocol B|NSFF [2]| 26.40 | 0.856 | 0.071 |
>     |Protocol C|DynIBaR [3]| 26.31 | 0.852 | 0.073 |
>
>     *We used this evaluation protocol in the main paper.
>
> 3. Considering that the focus of our work is on feed-forward reconstruction setting, we believe the reported benchmarks are sufficient to demonstrate that BTimer matches or surpasses existing per-scene optimization methods in quality while achieving several‑orders‑of‑magnitude faster runtime. Still, thank you so much for the suggestion.
>
> [1] Dynamic View Synthesis from Dynamic Monocular Video, ICCV 2021
>
> [2] Neural Scene Flow Fields for Space-Time View Synthesis of Dynamic Scenes, CVPR 2021
>
> [3] DynIBaR: Neural Dynamic Image-Based Rendering, CVPR 2023
>
> > **[W2]** Runtime performance
>
> Thank you for this valuable suggestion. To provide a more rigorous assessment of runtime, we have conducted an additional study varying both the number of context frames and the input resolution.
> | # Context frames | Resolution | Time (s) |
> |--|--|--|
> | 4 | 256x256 | 0.02s |
> |12| 256x256 | 0.15s |
> |12| 512x512 | 1.55s |
>
> The new results show that BTimer consistently remains in the second range (even at 512 × 512 resolution with 12 context frames), and is still several orders of magnitude faster than per‑scene optimisation pipelines. We will include these studies in the revised manuscript.
>
>
>
> > **[W3]** pixel-aligned prediction vs. XYZ prediction
>
> Thank you for raising this point. Canonical XYZ prediction is commonly used in object‑centric cases for being bounded (LGM, L4GM), however pixel-aligned prediction is popular in large unbounded scene for being easy to optimize (pixelSplat, GS-LRM). Applying canonical XYZ prediction to scene data is a valuable direction that we would like to explore in future work. And we will include the comparison discussion between them in our revised version.
>
>
>
> > **[Q1]** For a given in-the-wild video, what is the prescribed method for estimating camera poses and intrinsics?
>
> Given an in-the-wild video, we first apply per-frame Segment Anything Model to mask out pixel regions corresponding to the dynamic objects. The SAM model is prompted with several predefined categories of movable objects. In the meantime, we initialize the camera intrinsics using the GeoCalib model. We then apply DROID-SLAM on top of the video to estimate the camera poses, and the camera intrinsics is refined along with the poses in the global Bundle Adjustment process. We will release the full method and code for performing such camera annotations soon.
>
>
>
> > **[Q2]** Do the authors intend to release the source code and model checkpoints for re-implementation?
>
> Yes, the code and pretrained model will be released once accepted.

---

### Official Review · Reviewer_HRsg · 2025-07-02

**Clarity:** 4
**Significance:** 2
**Originality:** 2
**Rating:** 4
**Confidence:** 4

**Summary:**

The paper presents a method for novel view synthesis of dynamic scenes which is capable of showing the scene at a fixed point in time from unobserved viewpoints and also at interpolated time-points. For this the  paper proposes a ViT-based architecture, trained on a mixture of static and dynamic datasets, to infer a 3D Gaussian representation in a single forward pass, which can then be used to render views in real-time.

**Questions:**

**Major**
1. What is the contribution of the method beyond stitching existing components together?
2. Is there a specific reason to use different methods and different metrics for the two dataset? For example FPS is only reported for NVIDIA Dynamics Scenes (b) and SSIM is only reported for DyCheck (a).
3. How does the method compare to the current SoTA method DynIBaR?

**Minor**
1.  What is the exact meaning or intuition of "This is mainly caused by the inductive bias of pixel-aligned 3D Gaussian prediction" (L164)?
2. The paper uses a low resolution of 480 × 270 for for their quantitative evaluation and claims 150 FPS for NVIDIA Dynamics Scene (Tab. 1). How does this number change for high quality rendering (eg. Full-HD or 4K)?
3.  BTimer and 4D-GS rely on the same underlying representation. Can you give an explanation why BTimer is nearly 3x faster at rendering?

**Ethical Concerns:**

["NO or VERY MINOR ethics concerns only"]

**Final Justification:**

I thank the authors for the submitted rebuttal. After reviewing the rebuttal and discussion with the other reviewers, I am willing to increase the score from "Borderline reject" to "Borderline accept". For a higher score, the method would have required either a bigger technical contribution, instead of mostly relying on repurposed existing components, or more consistent results on the evaluated datasets.

**Limitations:**

The paper only contains a short high-level discussion of the method limitations and failure cases. There are several visual artefacts visible in the included in the supplementary video snippets which should have been discussed somewhere in the paper. Also all the presented scenes are reconstructed using forward facing cameras with small FoV and are only of a short duration (approx 5s), which should be included as additional limitations.

**Quality:**

3

**Strengths And Weaknesses:**

## Strengths

**1. Well written:** The paper is well written, easy to follow, and the figures are well made and are a good selection to give an overview of the method.

**2. Static scene evaluation:** The idea of also evaluating dynamic methods on a static setting is good. Since dynamic scenes are mostly made-up of static parts, this evaluation gives a good intuition how those parts will be reconstructed by the method.

**3. Number of qualitative examples:** The authors present a high number of qualitative examples from different datasets to give a good feel of the model performance.


## Weaknesses

**1. Contribution**: The biggest limitation of the paper seems to be the limited contribution. The core of the idea is a standard ViT architecture which is trained on a mixture of mostly common datasets with a standard image reconstruction loss. The difference here to existing static "feed-forward" reconstruction methods is the addition of a time-embedding (which is a basic positional encoding + linear layers) and a novel time enhancer (NTE) module. Anyhow, the idea behind NTE module is mostly taken from an existing method (LVSM, Jin et al. arXiv 2024) as mentioned by the authors. The used curriculum training is also pretty straight forward with the steps 1. coarse-to-fine on static scenes, 2. dynamic co-training, 3. extension of context window size to cover more complex deformations. Since the contribution on the side of the propose method is limited, more high-quality results (see below) would be required to justify a higher score for the paper.

**2. Mixed qualitative performance:** The quality of the rendered views varies between the selected scenes without a deeper discussion of this in the paper (see limitations review). It especially seems that the method has a hard time changing the perspective on the dynamic scene content in the bullet-time set-up. For example for the SORA scenes: The wave front in amalfi-coast_vid.mp4 seems to be floating above the ocean or the faces in grandma_birthday_vid.mp4 seem to deform with the selected viewpoint. Especially for high-fidelity scene contents such as the hand rolling the ball (iphone7.mp4 in the DyCheck dataset) the method seems to exhibit severe artefact's.

**3. Baseline selection**: As of 2025 DynIBaR (Li et al. CVPR 2023) seems to be still the current SoTA for high-quality dynamic novel view synthesis with a reported PSNR of >30 Db on Nvidia Dynamic Scenes. Arguably, the method takes substantially longer, but should be included as the max-quality baseline. Additionally, not all baselines are evaluated on both of the used datasets. For example the strongest method on the DyCheck dataset (T-NeRF) is not also evaluated on the NVIDIA dataset, where the authors then additionally report FPS.

---

> ### Author Rebuttal · Authors · 2025-07-30
>
> > **[W1,Q1]** What is the contribution of the method beyond stitching existing components together?
>
> We would like to clarify several points regarding our contributions:
> 1. **We propose the first feed‑forward dynamics scene reconstruction model with bullet‑time formulation.** Prior feed‑forward methods have been confined to static scenes due to (a) the challenge of modeling dynamics in a feed-forward manner and (b) the lack of large‑scale 4D supervision data.
>     - Our bullet‑time formulation extends well beyond a “standard ViT + time embedding” since it addresses both of the above issues in a novel manner, by training the network to aggregate context frames and predict the scene at an arbitrary target timestamp. This  implicitly encourages the model to become motion-aware and capture scene dynamics. To our knowledge, our formulation is the first feed-forward dynamic reconstruction method.
>     - More importantly, this formulation naturally unifies the static and dynamic reconstruction scenarios that allows us to pre-train our model on large amounts of static scene data and scales effectively across datasets, thus no longer being constrained by scarcity of 4D data. This scalability is key to generalization and performance, as shown in our results across static and dynamic benchmarks.
>
>     We believe this formulation is both conceptually novel and practically impactful, offering a significant step towards feed-forward real-time dynamic scene reconstruction.
>
> 2. Although our design is inspired by LVSM, **the NTE module differs in purpose and formulation**:
>     - NTE consumes the context frame, time embeddings for both context and target timestamps, and relative pose, whereas LVSM operates without time embedding.
>     - LVSM is designed for static reconstruction, while NTE is used within a dynamic‑reconstruction pipeline to predict missing intermediate frames under fast or sparse motion.
>     - NTE functions as an image‑space motion interpolator that complements the bullet‑timer, refining its 3DGS predictions and improving temporal consistency in challenging dynamic scenarios.
>
>     These differences make NTE a novel, task‑specific component rather than a direct reuse of LVSM.
>
> 3. **Our three‑stage curriculum is deliberately crafted to leverage abundant static data while gradually introducing motion complexity.** We also introduce interpolation losses which ensures a stable convergence of the model.
>
> 4. BTimer unites efficiency, robustness and fidelity. It delivers **orders of‑magnitude faster** inference than per‑scene optimization pipelines while **preserving comparable quality**. A single model generalises across a broad range of datasets with substantial distribution shifts, including real-world dynamic scenes, static scenes, AI-generated content (e.g., from Sora), and synthetic 4D environments (e.g., Kubric4D), whereas prior works train separate models for each domain.
>
> Together, we believe our contributions form a cohesive and non-trivial advancement in feed-forward dynamic scene reconstruction and address long-standing challenges in scalability, supervision, and real-time inference.
>
> > **[W2]** Mixed qualitative performance
>
> We appreciate the reviewer’s careful inspection of the qualitative videos and would like to clarify the observed artifacts:
> 1. Our model interpolates views by aggregating information across the context frames; consequently, its accuracy is proportional to the effective baseline and pose fidelity in the input. Most Sora clips contain only subtle camera motion and lack accurate pose estimates, making dynamic‑content extrapolation shown in bullet-time setup intrinsically ill‑posed. In scenes where the baseline is larger and the poses are more reliable (e.g., amalfi‑coast, cloud‑man, robot), the reconstructions are visibly sharper. We will highlight this dependency and add analysis in the revision.
> 2. Dycheck is a challenging dataset that features generic, high diverse dynamic scenes(MoSca [CVPR 2025]), where even many optimization‑based methods struggle. The `iphone 7` sequence, in particular, shows a hand moving much faster at close range, this is an edge case for many reconstruction techniques. Despite this, our feed‑forward model,
>     - outperforms several per‑scene optimisation approaches on this very clip (see the DyCheck evaluation page https://xiaoming-zhao.github.io/projects/pgdvs/videos/dycheck_eval/apple/index.html, where those approaches exhibit even more severe artefacts),
>     - matches or exceeds their quantitative scores across the entire DyCheck set,
>     - performs well on other hand sequences (`iphone4, 6, 8, 33`) with high fidelity.
>     We will add a brief discussion of this failure case and include comparisons to existing methods in revision.
> 3. As noted in Point 4 of our response to W1/Q1, BTimer shows strong robustness, efficiency, and reconstruction quality across diverse datasets.
>
> > **[W3.1, Q3]** Comparison with DynIBaR.
>
> 1. DynIBaR achieves high quality by performing hours of per‑scene optimisation (≈ 300 h per NSFF sequence) and by utilising additional supervision signals (optical flow + depth). In contrast, BTimer is a fully feed‑forward system: it reconstructs a scene within 1 second and is trained using only photometric loss. Given the vastly different computational budgets and data requirements, we belive that a head‑to‑head comparison is not strictly necessary for establishing our contribution.
> 2.  Nevertheless, we evaluate our method using DynIBaR’s protocol and compare it with the official DynIBaR results and the reproduction by PGDVS [ICLR 2024] (*) for reference.
>
>     | | PSNR↑ | SSIM↑ | LPIPS↓|
>     |--|--|--|--|
>     | DynIBaR | 30.86 | 0.957 | 0.027 |
>     | DynIBaR* | 29.35 | 0.934 | 0.0621 |
>     | BTimer | 26.31 | 0.852 | 0.0730 |
>
> > **[W3.2, Q2]** Not all baselines are evaluated on both of the used datasets.
>
> Thanks for this suggestion.
> 1. Baselines selection: we would like to clarify that we are not cherry-picking baselines. For each benchmark, we compare with methods that have official results on that benchmark, to ensure the comparison is rigorous and fair. This is exactly the same convention followed by previous dynamic‑scene reconstruction works (e.g., MoSca, CAT4D, PGDVS, Casual‑FVS, RoDynRF), where they likewise report different baseline sets across DyCheck and NVIDIA baselines for the same reason. If additional official results appear, we will incorporate them in the revision. Due to the limited rebuttal time, we were not able to re‑run T‑NeRF on the NVIDIA dataset (or DNeRF on DyCheck). However, we are happy to include these additional comparisons in the revised version.
> 2. Metric Comparison:
> For the same reason as baselines selections, to enable direct, fair comparisons with the published results of prior work, we follow each benchmark’s established protocol and adopt exactly the metrics that the existing baselines report.  On NVIDIA Dynamic Scenes, prior works (MoSca, Casual‑FVS, RoDynRF, MonoNeRF, DynNeRF) only list PSNR, LPIPS but not SSIM. We are happy to provide our SSIM score (`0.835`), however we didn’t find any official SSIM score from our compared baselines. FPS is reported in Casual‑FVS, so we adopt the same set. The reconstruction time for both two benchmarks have been reported in PGDVS. Anyhow, we will include complete metric comparison for those that are officially available in our revised version.
> 3. Considering that the focus of our work is on feed-forward reconstruction setting, we believe the reported benchmarks are already sufficient to demonstrate that BTimer matches or surpasses existing per-scene optimization methods in quality while achieving several‑orders‑of‑magnitude faster runtime.
>
> > **[Q4]** meaning of L164.
>
> Since the 3D position of each Gaussian is obtained through pixel-aligned unprojection (L121-122), this introduces a geometric bias that every Gaussian is anchored to a corresponding viewing ray of the input context frame. When we query an intermediate timestamp that has no corresponding context frame, the predictor struggles to predict the gaussian at that timestamp. In supplementary (L84) we train a variant of BTimer that removes this bias, which demonstrates that our BTimer formulation can successfully recover the 3D deformation and scene flow (Fig. S5-6 of supplementary).
>
> > **[Q5]** The paper uses a low resolution of 480 × 270 for for their quantitative evaluation and claims 150 FPS for NVIDIA Dynamics Scene (Tab. 1). How does this number change for high quality rendering (eg. Full-HD or 4K)?
>
> We additionally evaluate at 912x512 on Dynamic Scenes using the NSFF protocol. (Previous work evaluate at no higher than 288x540.) Due to limited rebuttal time, we test the existing 512x512 model without retraining a high resolution one, so quality drops somewhat. Inference rises from 0.44s to 4.48s due to the larger token count, yet it remains orders of magnitude faster than per-scene optimisation.
>
> | | PSNR↑ | SSIM↑ | LPIPS↓| Infer. Time↓ |
> |--|--|--|--|--|
> |480x270 | 26.40 | 0.856 | 0.0714 | 0.44s |
> |912x512| 24.59 | 0.788 | 0.1264 | 4.48s|
>
> > **[Q6]** BTimer and 4D-GS rely on the same underlying representation. Can you give an explanation why BTimer is nearly 3x faster at rendering?
>
> There are mainly two reasons:
> 1. 4D‑GS stores degree‑3 spherical‑harmonic (SH) coefficients for every Gaussian, while BTimer removes this overhead by storing a single RGB value, which cuts the computation and memory bandwidth per splat.
> 2. 4D-GS did not effectively reconstruct the scene from monocular video, hence more particles are used in order to overfit each view, causing the speed performance drop. Please note that we use their default configuration and directly trained on the NVIDIA benchmark, and both are tested on a single A100 GPU.

---

> > ### Author Response · Authors · 2025-08-04
> >
> > Dear reviewer `HRsg`,
> >
> > We would be grateful if you could let us know whether our rebuttal has addressed your concerns. Please don’t hesitate to reach out if you have any remaining questions or suggestions. We sincerely appreciate the time and effort you have dedicated to reviewing our paper.

---

> ### Comment · Reviewer_HRsg · 2025-08-05
>
> I thank the authors for their extensive rebuttal. The following questions have been sufficiently addressed: W2, W3.2/Q2, Q4, Q5, Q6.
>
> - **[W1/Q1]**: The rebuttal makes a strong case that the method has conceptual novelty, but the model still heavily relies on existing architectures and therefore has a limited technical novelty. Anyhow, I agree that there is a contribution in the integration and evaluation of the components.
> - **[W3.1/Q2]**: I am aware of the massive computationally effort it takes to utilize DynIBaR but I think it is importance to showcase the maximum possible performance for a posed problem to put the own performance into perspective. Therefore, I think it is important to include the results of DynIBaR since they are the actual SotA without additional restrictions such as limited computational budget/requirement of fast per-scene deployment. This does not mean that BTimer can claim SotA for feedforward reconstruction or a limited computational budget for inference.

---

> > ### Author Response · Authors · 2025-08-05
> >
> > We thank the reviewer for the thoughtful feedback. We would like to clarify that “feed-forward” reconstruction refers to the ability to directly infer novel scenes at test time **without** any per-scene optimization or retraining, which is a key distinction of our approach. Our method is, to our knowledge, the first feed-forward dynamic scene reconstruction model, capable of generalizing to unseen scenes with a single forward pass.
> >
> > While we appreciate the reviewer’s suggestion and are happy to include DynIBaR in our comparison, we emphasize that DynIBaR is not a feed-forward method—it requires ≈ 300 h of per-scene optimization and leverages additional supervision such as depth and optical flow. As such, DynIBaR can serve as a reference for maximum achievable quality under intensive computation and supervision, but it does not contradict our claim of achieving state-of-the-art performance among feed-forward dynamic reconstruction methods.

---

### Official Review · Reviewer_6QoZ · 2025-07-04

**Clarity:** 2
**Significance:** 2
**Originality:** 2
**Rating:** 4
**Confidence:** 4

**Summary:**

The paper introduces BTimer, a feed-forward model for real-time reconstruction and novel view synthesis of dynamic scenes from monocular videos, using a 3D Gaussian Splatting representation.

**Questions:**

1) The evaluation lacks comparisons with recent methods like CAT4D and MoSca, which outperform BTimer in dynamic reconstruction. Have the authors tested BTimer against these methods, and if not, why were they excluded given their relevance to feed-forward dynamic reconstruction?

2) The NTE module predicts intermediate frames for fast motions, but its contribution is unclear without ablation studies. Can the authors quantify the NTE’s impact on PSNR/SSIM/LPIPS and explain why it outperforms direct 3DGS prediction in fast-motion scenarios?

3) The model uses 12 context frames. How does BTimer scale to longer videos with more frames or higher temporal variability?

4) The NTE module predicts intermediate frames for fast motions, but its contribution is unclear without ablation studies. Can the authors quantify the NTE’s impact on PSNR/SSIM/LPIPS and explain why it outperforms direct 3DGS prediction in fast-motion scenarios?

**Ethical Concerns:**

["NO or VERY MINOR ethics concerns only"]

**Final Justification:**

I am satisfied with the authors’ responses and the evidence provided, and I am willing to increase my score to 4.

**Limitations:**

No limitations discussed. See Weaknesses.

**Paper Formatting Concerns:**

The paper generally follows the NeurIPS template.

**Quality:**

2

**Strengths And Weaknesses:**

Strengths:

1) The feed-forward approach enables real-time reconstruction, a significant advantage over optimization-based methods for applications like AR/VR and content creation.

2) The bullet-time formulation and NTE module are clearly described, with visualizations illustrating the pipeline effectively.

Weaknesses:

1) The contribution is incremental, as the bullet-time embedding and NTE module offer limited advancements over prior feed-forward models. The reliance on large-scale static and dynamic datasets raises concerns about generalizability to unseen scenarios without clear evidence of robustness.

2) The motivation for choosing timestamp embeddings over explicit motion modeling is weak.

---

> ### Author Rebuttal · Authors · 2025-07-30
>
> > **[W1.1]** Incremental Contribution of bullet-time embedding and NTE module
>
> We respectfully disagree with the assessment that our contributions are incremental.
>
> 1. Prior feed-forward models are limited to static scene reconstruction mainly due to two key challenges: (a). the complexity of modeling scene dynamics in a single forward pass, and (b). the scarcity of large-scale 4D dynamic supervision. Our proposed bullet-time formulation directly addresses both issues in a novel and elegant manner.
>
>     - the bullet-time design allows our model to predict the scene at an arbitrary bullet time by aggregating information from context frames. This training objective implicitly encourages the model to become motion-aware and capture scene dynamics. This is the first feed-forward dynamic reconstruction formulation that has never been explored before.
>
>     - More importantly, this formulation naturally unifies the static and dynamic reconstruction scenarios that allows us to pre-train our model on large amounts of static scene data and scales effectively across datasets. This scalability is key to generalization and performance, as shown in our results across static and dynamic benchmarks.
>
>     We believe this formulation is both conceptually novel and practically impactful, offering a significant step towards feed-forward real-time dynamic scene reconstruction.
>
> 2. We propose the NTE module to further enhance reconstruction quality, particularly under complex and fast motions. It introduces a novel design by conditioning on the context frame, interpolated timestamp embedding, and relative pose. This design complements the bullet-time prediction and can be seamlessly integrated into our framework.
>
> 3. Moreover, we also introduce curriculum training at scale and interpolation supervision, which are critical to ensuring stable convergence. As demonstrated in our ablation studies, each of these components contributes meaningfully to the overall performance of the model.
>
> Taken together, we believe our contributions form a cohesive and non-trivial advancement in feed-forward dynamic scene reconstruction and address long-standing limitations around scalability, supervision, and real-time inference.
>
>
>
> > **[W1.2]** Reliance on large-scale static and dynamic datasets raises concerns about generalizability.
>
> 1. To rigorously evaluate the generalization ability of our model, we conduct extensive experiments across a broad range of datasets, including real-world dynamic scenes, static scenes, AI-generated content (e.g., from Sora), and synthetic 4D environments (e.g., Kubric4D). These datasets exhibit substantial domain gaps, yet our model maintains consistent and strong performance, achieving results competitive with per-scene optimization methods and significantly outperforming previous feed-forward baselines. We believe this provides compelling empirical evidence of robustness and cross-domain generalization.
>
> 2. To be the same with most recent feed-forward reconstruction methods (e.g., GS-LRM [1], LGM [2], DUSt3R [3], VGGT [4]), our model leverages large-scale datasets to enhance generalization capacity. This reliance is not a limitation, but a necessary component for real-time, generalizable scene understanding at scale. We emphasize that our design further amplifies the benefits of such pretraining by unifying static and dynamic reconstruction within a single formulation.
>
> 3.  We further include additional experiments to demonstrate transfer performance across datasets. We simplify the training setting and only use RE10K in the static stage and a mix of Kubric4D and RE10K in the dynamic stage. The model is evaluated on the Dynamic Scenes dataset. We highlight that Kubric4D is a synthetic dataset with rigid motion, while Dynamic Scene is a real-world dataset with articulated and deformable motion. Despite the significant domain difference, training on Kubric4D substantially improves all metrics on Dynamic Scenes, which shows that the dynamic representation learned can be effectively transferred across very different datasets.
>
>
>     | | PSNR↑ | SSIM↑ | LPIPS↓|
>     |--|--|--|--|
>     |RE10K| 21.62 |  0.724 | 0.257 |
>     |RE10K + Kubric4D | **23.61** | **0.755** |  **0.214**|
>
>
> [1] GS-LRM: Large Reconstruction Model for 3D Gaussian Splatting, ECCV 2024
>
> [2] LGM: Large Multi-View Gaussian Model for High-Resolution 3D Content Creation, ECCV 2024
>
> [3] DUSt3R: Geometric 3D Vision Made Easy, CVPR 2025
>
> [4] VGGT: Visual Geometry Grounded Transformer, CVPR 2025
>
>
>
> > **[W2]** Weak motivation of timestamp embeddings over explicit motion modeling
>
> Timestamp embeddings offer important advantages over explicit motion modeling.
>
> 1. It allows our model to capture flexible and complex dynamics, such as deformation with topological changes, and complex natural phenomena such as fire, waves, smoke, and dust. This is evident in our supplementary visualizations on SORA and DAVIS datasets.
>
> 2. Despite using implicit time conditioning, our model can still recover decent 3D deformation and scene flows as evidenced in Figure S5 and S6 of the supplementary, where the dynamic structure of the scenes is clearly recovered. This indicates that the model learns meaningful motion representations from time embeddings, even without explicit motion supervision.
>
> 3. Our approach aligns with several prior works that successfully use implicit time modeling for dynamic scenes, including ST-NeRF [5], DyNeRF [6], and HexPlane [7].
>
>
> [5] Space-Time Neural Irradiance Fields for Free-Viewpoint Video, CVPR 2021
>
> [6] DyNeRF: Neural 3D Video Synthesis from Multi-view Video, CVPR 2022
>
> [7] HexPlane: A Fast Representation for Dynamic Scenes, CVPR 2023
>
>
>
> > **[Q1]** Compared with CAT4D and MoSca
>
> 1. Our primary focus is on fully feed-forward dynamic scene reconstruction, while both CAT4D and MoSca rely on lengthy per-scene optimization to achieve 4D reconstruction. CAT4D’s code and pretrained weights are not available, which precludes a broader empirical comparison. And, in MoSca’s case, it requires additional 2‑D tracks and depth supervision for optimization. Hence, a direct comparison is not strictly required.
>
> 2. Notwithstanding the differing problem settings and supervisions, we make a comparison on the NSFF benchmark, the only common dataset on which CAT4D and MoSca report results. Using their own reported scores, our method significantly surpasses CAT4D and is on par with MoSca, while our model is trained with only photometric loss, and runs in a feed-forward manner.
>
>
>     | | PSNR↑ | SSIM↑ | LPIPS↓|
>     |--|--|--|--|
>     | MoSca | 26.72 | - | 0.070 |
>     | CAT4D | 21.97 | 0.683 | 0.121 |
>     | BTimer | **25.82** | **0.835** | **0.086** |
>
>
>
> > **[Q2.1]** Quantitative effect of NTE module
>
> 1. NTE is designed to improve reconstructions when the temporal frame is sparse or motion is fast, by refining predictions at intermediate timestamps. On standard dynamic benchmarks such as Dynamic Scenes, frames are dense and motion magnitude is limited; the standard protocal also does not evaluate in-between timestamps.
> 2. We have shown qualitative ablation in Fig6. To provide quantitative evaluation, we additionally follow the interpolation evaluation protocol in NSFF by extracting every other frame from the original Dynamic Scenes dataset. `w/ GT` means using the ground truth in-between frames which shows the upper bound of the interpolation model. NTE effectively improves on PSNR and SSIM. Note that the slight drop on LPIPS is expected, since perceptual metrics will worsen with interpolation [8].
>
>     | | PSNR↑ | SSIM↑ | LPIPS↓|
>     |--|--|--|--|
>     | w/o NTE | 25.73 | 0.844 | 0.0786 |
>     | w/ NTE | 25.97 (+0.24) |  0.846 (+0.02) | 0.0791 (+5e-4) |
>     | w/ GT | 26.29 (+0.56) | 0.851 (+0.07)| 0.0734 (-5.2e-3)|
>
>
>     [8] The Perception-Distortion Tradeoff, CVPR 2018
>
>
>
> > **[Q2.2]** Reason for outperforming direct 3DGS prediction in fast-motion scenarios.
>
> As noted in L160–167, when temporal frames are sparse or motion is fast, the frame corresponding to an intermediate timestamp is absent from the context. Due to the inductive bias of pixel‑aligned 3D‑Gaussian predictor, the model struggles to produce smooth transitions. Since the NTE module is not subject to this 3D bias, it synthesizes the missing intermediate view accurately; we then pass this NTE output to BTimer for the final reconstruction.
>
>
> > **[Q3]** The model uses 12 context frames. How does BTimer scale to longer videos with more frames or higher temporal variability?
>
> BTimer can accommodate lengthy or highly dynamic sequences through several practical strategies:
>
> 1. At test time we apply BTimer in a sliding window manner; thus, memory usage and latency remain constant, regardless of the total video length.
> 2. We can subsample representative key frames and pair them with the context frames corresponding to the target timestamp as input to the model. This reduces computational load while preserving global coherence.
> 3. The architecture readily accepts more frames. By replacing the standard ViT blocks with recent Mamba‑2 blocks, as adopted in Long‑LRM (ICCV 2025), we can process substantially more tokens efficiently.
>
>
> Hence, these strategies allow BTimer to scale gracefully to longer videos and higher temporal variability without requiring fundamental architectural changes.

---

> > ### Author Response · Authors · 2025-08-04
> >
> > Dear reviewer `6QoZ`,
> >
> > We would be grateful if you could let us know whether our rebuttal has addressed your concerns. Please don’t hesitate to reach out if you have any remaining questions or suggestions. We sincerely appreciate the time and effort you have dedicated to reviewing our paper.

---

> > ### Comment · Reviewer_6QoZ · 2025-08-04
> >
> > Thank you for your detailed rebuttal. You have addressed many of the concerns raised during the review process.
> >
> > However, I would like to raise a fundamental question for deeper reflection: Is the proposed "bullet-time formulation" truly a novel paradigm for dynamic scene reconstruction, or is it more accurately described as an effective engineering integration of timestamp embeddings with a feed-forward 3DGS prediction framework? While the formulation enables reconstruction at arbitrary timestamps and leverages both static and dynamic data through a unified pipeline, similar time-conditioned architectures have been explored in prior works on dynamic scenes (e.g., T-NeRF, HexPlane). The core mechanism—aggregating context frames via attention, conditioned on target time embeddings—appears to be a natural extension of existing feed-forward models rather than a fundamental departure in how motion is modeled. Furthermore, the NTE module, while empirically beneficial, functions primarily as a 2D video interpolator, enhancing input conditioning but not inherently modeling 3D motion dynamics. As such, the claim of being the "first motion-aware feed-forward model" may overstate the conceptual novelty, as the motion awareness seems to emerge implicitly from data rather than from an explicit, structured dynamic prior.
> >
> > Overall, I am satisfied with the authors’ responses and the evidence provided, and I am willing to increase my score to 4 if my concerns have been adequately resolved.

---

> > > ### Author Response · Authors · 2025-08-04
> > >
> > > Thank you for your follow-up and your willingness to raise your score. We would like to clarify the key distinctions:
> > >
> > > 1. Our bullet-time has a fundamentally different formulation from prior time-conditioned models like T-NeRF and HexPlane. Those works predict RGB values conditioned on camera viewpoint and target timestamp, but overfit to one single scene. In contrast, our method works in a feed-forward manner, and directly predicts the 3D Gaussians at arbitrary target timestamps, conditioned on RGB context frames, viewpoints, their *context* timestamps, as well as the *target* timestamp. This formulation enables temporal reasoning across frames by training from large-scale datasets, and allows for "bullet-time" effects and consistent dynamic scene reconstruction in a fully feed-forward manner. This capacity is not addressed by existing methods.
> > > 2. NTE is not merely a video interpolator, particularly, it is viewpoint-aware (hence the output is more 3D consistent) and supports novel view synthesis. This module complements our bullet-timer by refining 3DGS predictions in fast-motion scenarios and enhancing temporal consistency in challenging dynamic scenes, which can be seamlessly integrated into our feed-forward pipeline.
> > > 3. Our claim of "first motion-aware feed-forward model" is grounded in both capability and formulation, and we believe it is well-justified despite the use of implicit dynamics.
> > >    - Our model is motion-aware because:
> > >
> > >      (a). it is able to aggregate dynamic context input across multiple viewpoints and timestamps to build accurate 3D representations at arbitrary target times.
> > >
> > >      (b). our model learns to recover meaningful 3D deformation and scene flow, as evidenced by Figure S5 and S6 in the supplementary, even without explicit motion supervision.
> > >
> > >    - Modeling dynamics implicitly provides advantage over explicit modeling:
> > >
> > >      (a). Modeling dynamics implicitly allows our model to capture flexible and complex dynamics, such as deformation with topological changes, and complex natural phenomena such as fire, waves, smoke, and dust. This is evident in our supplementary visualizations on SORA and DAVIS datasets.
> > >
> > >      (b). While explicit-motion models (e.g., 4DGS [CVPR’24] built on HexPlane) can successfully overfit individual 4D sequences, we experimentally find that they struggle to learn stable deformation fields in large-scale, feed-forward training settings, as also observed in L4GM [NeurIPS’24]. In contrast, our implicit modeling approach leads to more stable training and generalization well across scenes.

---

> > > ### Author Response · Authors · 2025-08-07
> > >
> > > Dear Reviewer 6QoZ,
> > >
> > > We sincerely appreciate your engagement and the time invested in this discussion phase. If you have further inquiries, we would be honored to provide clarification to the best of our ability.
> > >
> > > If you determine that all concerns have been adequately addressed, we would be deeply honored if you could kindly reconsider your evaluation. Thank you again for your devotion to the review.
> > >
> > > Best regards,
> > > The Authors of Paper #17189

---

### Decision · Program_Chairs · 2025-09-17

**Decision:**

Accept (poster)

**Comment:**

This submission introduces a feed-forward solution for dynamic scene reconstruction and novel view synthesis. The reviewers provided positive feedback (with three borderline accepts and one accept). The AC agrees with the reviewers that the problem addressed by the submission is important and that the technical solution is sound.
The AC recommends accepting the paper, subject to significant revisions, including:
- 1. The AC finds that a highly relevant discussion on feed-forward 4D reconstruction is missing. Please incorporate such a discussion in the revision. A useful reference is Section 3.4 of Advances in Feed-Forward 3D Reconstruction and View Synthesis: A Survey.
- 2. Based on the comparison results in Table 1 and the additional results provided in the rebuttal with DynIBaR, the authors should moderate their state-of-the-art performance claims in the abstract. The authors are also encouraged to include comparisons with NeRFPlayer and Shape of Motion.
- 3. Please include an inference speed performance table and explicitly state the image resolution when referring to the reported 150 ms per frame.
- 4. Improve the clarity of the misleading points pointed out by the reviewers.